# Selective synthesis of tightly- and loosely-twisted metallomacrocycle isomers towards precise control of helicity inversion motion

Tomoki Nakajima[1], Shohei Tashiro [1] ✉, Masahiro Ehara [2] &
Mitsuhiko Shionoya [1] ✉

Molecular twist is a characteristic component of molecular machines. Selectively synthesising isomers with different modes of twisting and controlling their motion such as helicity inversion is an essential challenge for achieving more advanced molecular systems. Here we report a strategy to control the inversion kinetics: the kinetically selective synthesis of tightly- and loosely-twisted isomers of a trinuclear $Pd^{II}$-macrocycle and their markedly different molecular behaviours. The loosely-twisted isomers smoothly invert between (P)- and (M)-helicity at a rate of $3.31\ s^{-1}$, while the helicity inversion of the tightly-twisted isomers is undetectable but rather relaxes to the loosely-twisted isomers. This critical difference between these two isomers is explained by the presence or absence of an absolute configuration inversion of the nitrogen atoms of the macrocyclic amine ligand. Strategies to control the helicity inversion and structural loosening motions by the mode of twisting offer future possibilities for the design of molecular machines.

One of ultimate goals in the chemistry of molecular machines is to reproduce motions in the macroscopic world at the level of molecules or their aggregates. A number of excellent examples of molecular machines have been reported to control a variety of motions, including rotation[1–5] and translation[6–10]. However, controlling motion elements such as directionality[3,11], speed[4,10] and frequency (ON/OFF)[9,12] is still a major challenge. For example, Stoddart et al. controlled the translational motion of rings shuttling along the axles by redox or acid/base[6]. Feringa et al. reported unidirectional molecular rotors based on photo- and thermal-isomerisation of olefins[11]. In addition to these motions, twisting motion and its inversion motion have recently attracted much attention in terms of motion complexity and asymmetry, leading to advanced molecular machines and chiral memory[13–17].

We intuitively understand that the twisting behaviours are highly dependent on the mode of twisting. For instance, a loosely-twisted object will unwind easily, but a very tightly-twisted object may not. Can such movements be reproduced in the nanoscale world? Inspired by biomacromolecules, chemists have synthesised twisted molecules such as helical polymers[18–20], helicenes[21,22] and twisted macrocycles[23–25],

and in some of these examples, the design and control of their twisting motion are being studied. For instance, the rate of helicity inversion was controlled by exploiting the kinetic properties of coordination bonds in twisted metal complexes of synthetic peptides[26], macrocycles[27] and cryptands[28]. Another role of metal coordination is to dynamically fix the absolute configuration of the coordinating atoms, including the amine nitrogen atoms, which is usually immediately reversed. Thus, selective synthesis of metal complexes with different modes of twisting resulting in different configurations and/or conformation, or twisted isomers is a promising strategy to control the inversion motion without changing the chemical composition, but such isomers are limited to a few examples[29–32]. Thus, it is still challenging to selectively synthesise such twisted isomers.

Molecular motion is described by a combination of rotational and translational motions, but various modes are possible depending on the shape and size of the molecule itself, the flexibility of the structure derived from the bonding modes and the environment surrounding the molecule. Molecular twist is a mode included in many molecular motions of molecular machines. Selective synthesis of isomers with

[1]Department of Chemistry, Graduate School of Science, The University of Tokyo, Tokyo, Japan. [2]Research Center for Computational Science, Institute for Molecular Science, Myodaiji, Okazaki, Aichi, Japan. ✉e-mail: tashiro@chem.s.u-tokyo.ac.jp; shionoya@chem.s.u-tokyo.ac.jp

different mode of twisting and control of their motion such as inversion, is an indispensable task to realise more advanced molecular systems.

Herein we report the selective synthesis of two twisted isomers of a trinuclear $Pd^{II}$-macrocycle with a tightly- or loosely-twisted skeleton (Fig. 1). It is particularly important to emphasise that these two isomers have markedly different rates of helicity inversion depending on the mode of twisting with different absolute configurations of the diamine moieties locked by the metal ions. The loosely-twisted isomer exhibited rapid helicity inversion, whereas the helicity inversion in the tightly-twisted isomer was actually undetectable because the inversion process requires absolute configuration inversion of the nitrogen atoms. In other words, the helicity inversion of the twisted macrocycle is configurationally inhibited by twisting more tightly. Thus, this result is an excellent example of twisting motion controlled by the mode of twisting of a single chiral molecule with coordinating atoms of different configurations.

## Results

### Selective synthesis of two twisted isomers, $1_{tight}$ and $1_{loose}$ (Fig. 2)

Previously, our group synthesised helically-twisted $Pd^{II}$-macrocycles, $[Pd_3LCl_6]$, from an achiral macrocyclic hexaamine ligand **L** and 3 equiv. of $[PdCl_2(CH_3CN)_2]$ in $CH_2Cl_2$:DMSO = 9/1 (v/v)[33]. The $C_3$-symmetric macrocyclic skeleton with three $Pd^{II}$ centres on the same side was helically twisted by intramolecular C-H⋯π interactions (Fig. 3i, j). This complex exhibits helicity inversion between the (*P*)- and (*M*)-enantiomers with an inversion rate of 14 s$^{-1}$ at 292 K in $CD_2Cl_2$:DMSO-$d_6$ = 9/1 (v/v). In this study, we aimed to significantly modulate the rate of helicity inversion by replacing all two chloride ligands on the three $Pd^{II}$ centres with 4,4′-di-*tert*-butyl-2,2′-bipyridine (*t*Bu$_2$bpy) ligands.

First, we attempted to synthesise a twisted $Pd^{II}$-macrocycle with Pd(*t*Bu$_2$bpy) moieties in a manner similar to our previous study[33]. Ligand **L** was reacted with 3.2 equiv. of $[Pd(^tBu_2bpy)(OH_2)_2](OTf)_2$ in $CH_2Cl_2$ at room temperature for 4 h. After removal of the solvent, the residue was recrystallised to afford colourless crystals $1_{tight}$ in 33% yield (Fig. 2). The composition was determined by elemental analysis to be $[Pd_3L(^tBu_2bpy)_3](OTf)_6·(H_2O)_{4.9}·(Et_2O)_{0.15}$. Single-crystal X-ray diffraction (ScXRD) analysis revealed that $[Pd_3L(^tBu_2bpy)_3](OTf)_6$ has a $C_3$-symmetric twisted structure (Fig. 3a–c), and the twisting

macrocyclic structure was different from that of $[Pd_3LCl_6]$ (Fig. 3j). Specifically, the three *ortho*-phenylenediamine moieties folded inside the macrocycle to form a tightly-twisted skeleton, and this complex $[Pd_3L(^tBu_2bpy)_3](OTf)_6$ is referred to here as the tightly-twisted isomer ($1_{tight}$): the other loosely-twisted isomer ($1_{loose}$) will be discussed below (the definition of the twisted isomers; see the Supplementary Section 2.4). This isomer $1_{tight}$ has (*P*)- or (*M*)-helicity, defined by the direction from the *para*-phenylene rings toward the inner amine protons (Fig. 2), and both enantiomers crystallised as a racemate. Unlike $[Pd_3LCl_6]$, $1_{tight}$ had three amine protons facing outward of the macrocycle and the other three facing inward, forming hydrogen bonds with one triflate ion in the cavity. The absolute configurations of the six nitrogen atoms, the chiral centres in the (*P*)- and (*M*)-enantiomers, were therefore *all-R*- and *all-S*-configuration, respectively. Intramolecular C-H⋯Pd interactions between one of the protons of *para*-phenylene and the Pd centre were suggested by the H⋯Pd distance (2.74 Å) and the C-H⋯Pd angle (120°) as one factor stabilising the tightly-twisted structure. These are consistent with typical anagostic interactions (2.3–2.9 Å, 110–170°)[34]. To confirm the structure in solution, the crystals were then dissolved in acetone or dichloromethane and analysed by 1D $^1H$ and $^{19}F$ NMR spectroscopies, 2D NMR spectroscopies ($^1H$–$^1H$ COSY and ROESY), and high resolution-electrospray ionization time-of-flight (HR-ESI-TOF) mass spectrometry ($m/z = 1100.2498$ as $[Pd_3(H_{-1}L)(^tBu_2bpy)_3(OTf)_3]^{2+}$) (Supplementary Figs. 4–16). The $^{19}F$ NMR spectrum showed two separate triflate signals, one of which was assigned to a triflate incorporated within the macrocycle (Supplementary Fig. 6). Two sets of diastereotopic methylene proton signals ($H_{c-f}$) were observed in the $^1H$ NMR spectrum in acetone-$d_6$, indicating that the structure is chiral (Fig. 3a, d). Notably, the downfield shift of one *para*-phenylene signal ($H_l$) to 9.8 ppm even at 300 K suggested the presence of a C-H⋯Pd interaction (Supplementary Fig. 14). Moreover, an *ortho*-phenylenediamine proton signal ($H_g$) was highly upfield shifted to 4.9 ppm due to the shielding effect from the adjacent *ortho*-phenylenediamine moieties clustered inside the macrocycle. These results suggest that the tightly-twisted structure is maintained in solution. The interactions suggested by ScXRD and NMR analyses were well supported by natural bond orbital (NBO) and noncovalent interaction (NCI) plot analyses after optimising the geometry with density functional theory (DFT) calculations (Supplementary Figs. 118–120).

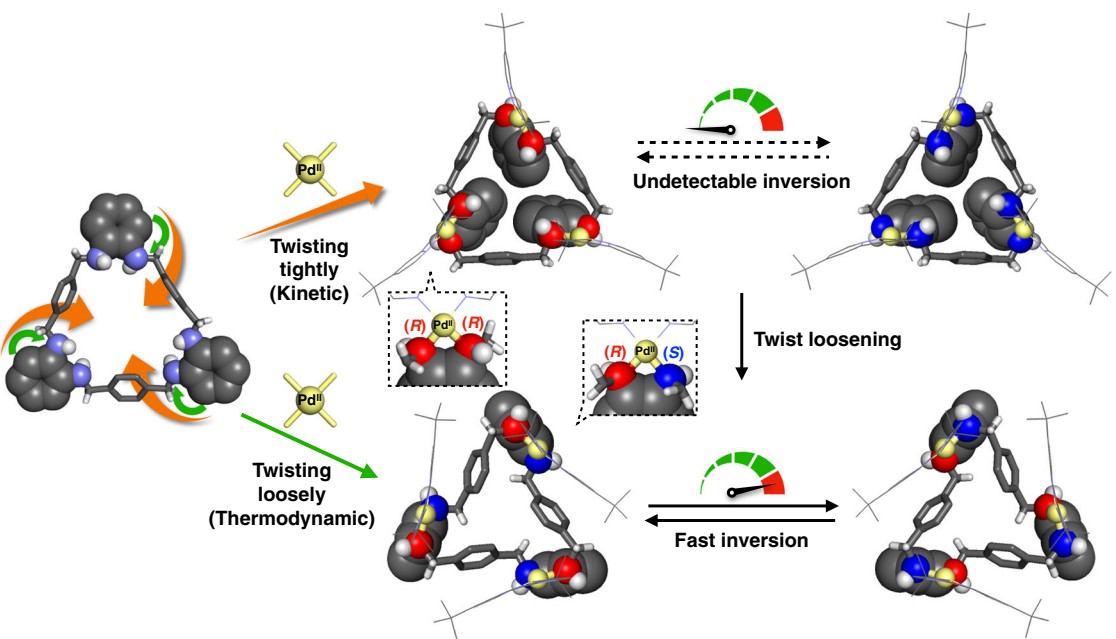

**Fig. 1 | The concept of molecular helicity inversion controlled by twisting mode due to differences in the absolute configurations of the diamine moieties locked by the metal ions.** Colour: Pd yellow, C black, H white, N Purple (non-coordinated), red (*R*-configuration) and blue (*S*-configuration).

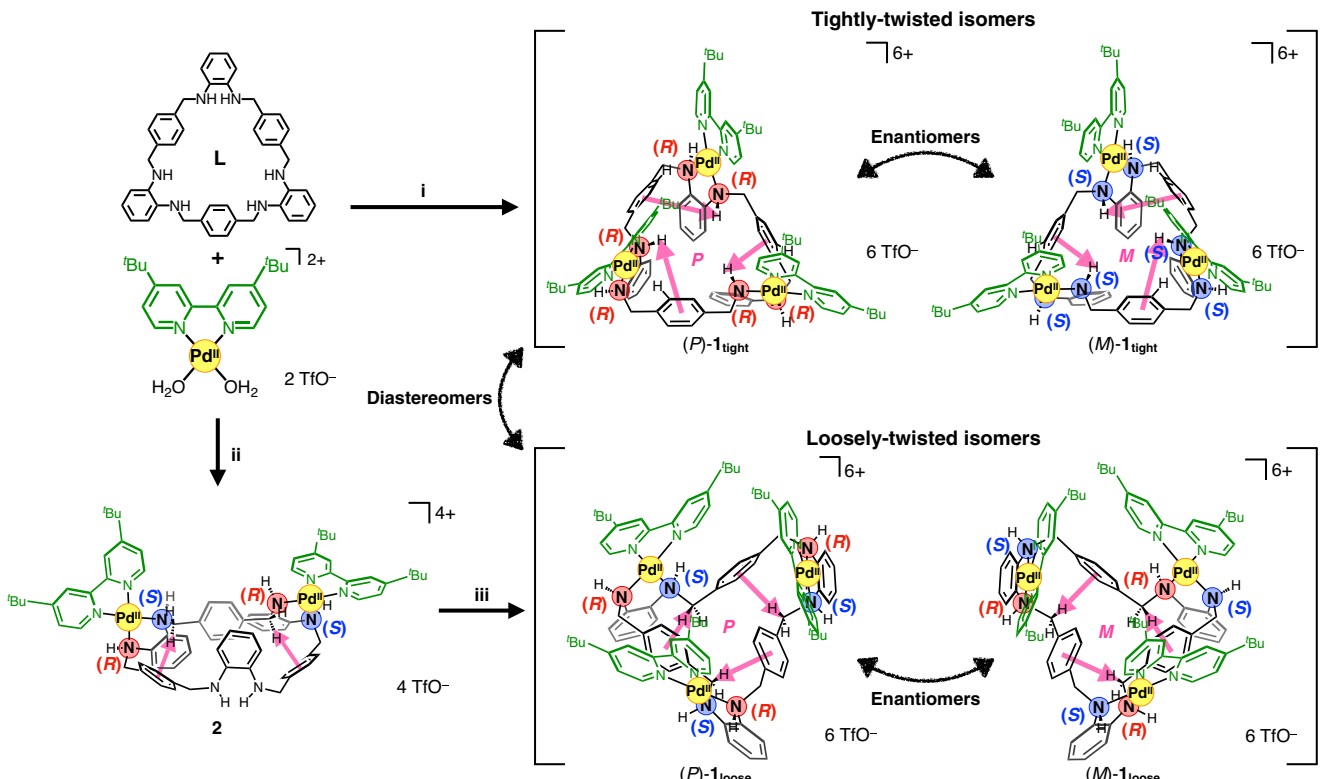

**Fig. 2 | Synthesis of two isomeric Pd^II complexes, 1_tight and 1_loose, and the absolute configuration of the amine nitrogen atoms.** i CH$_2$Cl$_2$, rt, 4 h, 33%. ii CHCl$_3$, rt, 3 h, 64%. iii [Pd($^t$Bu$_2$bpy)(OH$_2$)$_2$](OTf)$_2$, CH$_2$Cl$_2$, rt for 3 h then reflux for 1.5 h, 49% (31% in two steps).

To synthesise another isomeric trinuclear Pd^II-macrocycle with a skeleton similar to [Pd$_3$LCl$_6$], we optimised the reaction conditions and found that a dinuclear Pd^II-macrocycle is a key intermediate for the selective synthesis of the other isomers (Fig. 2). When ligand **L** was reacted with 1.6 equiv. of [Pd($^t$Bu$_2$bpy)(OH$_2$)$_2$](OTf)$_2$ in CHCl$_3$ at room temperature, a dinuclear [Pd$_2$L($^t$Bu$_2$bpy)$_2$](OTf)$_4$ (**2**) complex precipitated. Its *meso*-twisted skeleton with two intramolecular C-H···π interactions was deduced from the crystal structure of a Pt^II-analogue, 1D and 2D NMR and HR-ESI-TOF mass analyses (*m/z* = 689.2654 as [Pd$_2$(H$_{-2}$L)($^t$Bu$_2$bpy)$_2$]$^{2+}$) (Supplementary Figs. 34–45). Dinuclear complex **2** was then reacted again with 1.2 equiv. of [Pd($^t$Bu$_2$bpy)(OH$_2$)$_2$](OTf)$_2$ in CH$_2$Cl$_2$, and the product was recrystallised to afford colourless crystals **1_loose**, [Pd$_3$L($^t$Bu$_2$bpy)$_3$](OTf)$_6$·(H$_2$O)$_4$, in 31% total yield. ScXRD analysis revealed that the crystals were composed of another $C_3$-symmetric trinuclear complex with a twisted skeleton similar to [Pd$_3$LCl$_6$] (Fig. 3e–g, i, j). Unlike **1_tight**, the three *ortho*-phenylenediamine moieties were not folded much and located outside the macrocyclic structure, instead, the inside was filled with *para*-phenylene and three methylene moieties, forming C-H···π interactions between them. Thus, this complex [Pd$_3$L($^t$Bu$_2$bpy)$_3$](OTf)$_6$ (**1_loose**) can be regarded as a loosely-twisted isomer of **1_tight**. This isomeric complex also formed racemic crystals consisting of (*P*)- and (*M*)-enantiomers. In contrast to **1_tight**, all six amine protons of **1_loose** were located outside of the macrocyclic structure, resulting in an alternating *R*- and *S*-absolute configuration of nitrogen atoms. Therefore, helicity inversion between the (*P*)- and (*M*)-**1_loose** preserves the alternate absolute configurations (*alt-R/S*) of the nitrogen atoms. One triflate ion was found to be disordered in the space surrounded by three bipyridine portions. Analyses of **1_loose** dissolved in acetone by 1D and 2D NMR spectroscopies and HR-ESI-TOF mass spectrometry (*m/z* = 1100.2410 as [Pd$_3$(H$_{-1}$L)($^t$Bu$_2$bpy)$_3$(OTf)$_3$]$^{2+}$) revealed that the structure of **1_loose** in solution is consistent with the crystal structure (Supplementary Figs. 19–29). For instance, one methylene signal (H$_c$) was upfield shifted to 1.9 ppm in ¹H

NMR, which supported the C-H···π interactions observed in the crystal structure (Fig. 3f, g). This interaction was also supported by NBO and NCI plot analyses after geometry optimisation using DFT calculation (Supplementary Figs. 126 and 127).

**1_loose** was stable in acetone-$d_6$ at room temperature for two weeks, as evidenced by ¹H NMR analysis (Supplementary Fig. 32). In contrast, **1_tight** was less stable and slowly isomerised to **1_loose** in acetone-$d_6$ while also producing non-assignable by-products. The isomerisation rate of **1_tight** to **1_loose** in acetone-$d_6$ at 293 K was estimated to be $(5.7 \pm 0.4) \times 10^{-6}$ s$^{-1}$ by the time course ¹H NMR analysis, assuming the isomerisation to be a pseudo first order reaction (Supplementary Section 3). This result indicates that **1_tight** and **1_loose** are the kinetic and thermodynamic products in acetone, respectively, and is consistent with the fact that **1_tight** is obtained kinetically under mild conditions in CH$_2$Cl$_2$. As mentioned above, dinuclear complex **2**, which preferentially precipitated when reacted with less than 2 equiv. of Pd^II salts in CHCl$_3$, is an important intermediate for the selective synthesis of **1_loose**. This result is consistent with the fact that the absolute configuration of the four Pd^II-coordinated amine nitrogen atoms in **2** is the same as that of **1_loose** and no configurational change is required when **2** is converted to **1_loose**. In contrast, to convert to **1_tight** with an *all-R-* or *all-S*-configuration, **2** requires a configurational inversion of the two Pd^II-coordinated amine nitrogen atoms. Thus, the conversion from **2** to **1_tight** is kinetically undesirable, resulting in a selective conversion to **1_loose** (Fig. 2).

The difference in the absolute configuration of the Pd^II-coordinated amine nitrogen atoms is thus a notable structural difference between **1_tight** and **1_loose**. It is noteworthy that the two diastereomers can also be regarded as *in/out*-isomers if we focus on the nitrogen-containing cyclic structures, which is usually applied to bridged bicyclic compounds[35]. Here, the two nitrogen atoms of one *ortho*-phenylenediamine moiety are the bridgehead atoms of the bicyclic

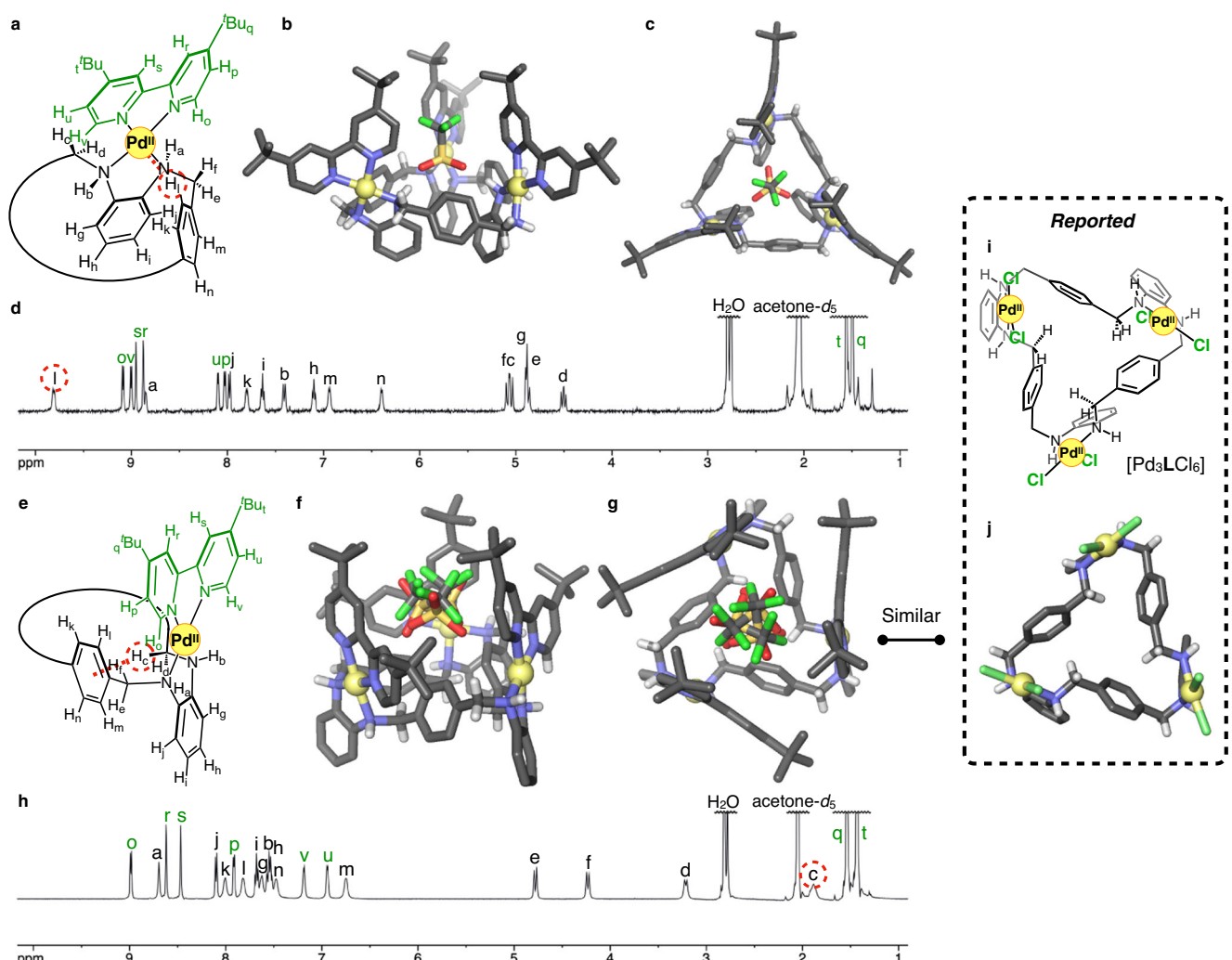

**Fig. 3 | Characterisation of the two twisted isomers of 1. a** Substructural formula of **1**$_\text{tight}$. **b, c** Side and top views of the crystal structure of **1**$_\text{tight}$ with one triflate incorporated via multipoint hydrogen bonding. Hydrogen atoms except amine and methylene moieties are omitted for clarity. **d** $^1$H NMR spectrum (500 MHz, acetone-$d_6$, 300 K) of **1**$_\text{tight}$. **e** Substructural formula of **1**$_\text{loose}$. **f, g** Side and top views of the

crystal structure of **1**$_\text{loose}$ with one triflate disordered in the inner space. Hydrogen atoms except amine and methylene moieties are omitted for clarity. **h** $^1$H NMR spectrum (500 MHz, acetone-$d_6$, 300 K) of **1**$_\text{loose}$. **i, j** Structural formula and the reported crystal structure (top view) of [Pd$_3$**L**Cl$_6$][33].

structure bridged via Pd$^\text{II}$. According to the definition of *in/out*-isomers, **1**$_\text{tight}$ and **1**$_\text{loose}$ correspond to the *in,out*- and *out,out*-isomers, respectively, from the direction of the N-H moieties.

## Estimation of helicity inversion rate by exchange spectroscopy (EXSY)

The inversion rates of **1**$_\text{loose}$ and **1**$_\text{tight}$ were then investigated. The helicity inversion rate ($k$) between ($P$)- and ($M$)-**1**$_\text{loose}$ was evaluated using EXSY with varying mixing time. The rate constant ($k$) was calculated using the ratio of integrations of chemical exchange signals (e.g., H$_\text{o}$ and H$_\text{v}$ for different bipyridine moieties) produced by helicity inversion, which was $3.31 \pm 0.02\,\text{s}^{-1}$ at 300 K in acetone-$d_6$ (Fig. 4a–c, Supplementary Sections 4.1, 4.3). In the EXSY analysis, the rate constant ($k$) obtained was defined as the sum of the rate constants of the helicity inversion from ($P$)- to ($M$)-**1**$_\text{loose}$ ($k_\text{PM}$) and from ($M$)- to ($P$)-**1**$_\text{loose}$ ($k_\text{MP}$) (Fig. 4a)[36]. The Gibbs free energy ($\Delta G^\ddagger$), enthalpy ($\Delta H^\ddagger$) and entropy ($\Delta S^\ddagger$) of activation at 300 K were estimated to be $70.6 \pm 1.3\,\text{kJ/mol}$, $86.1 \pm 0.9\,\text{kJ/mol}$ and $51.9 \pm 3.1\,\text{J/(mol·K)}$, respectively, by the Eyring plot based on the inversion rate at temperatures varying in the range from 280 to 310 K (Supplementary Figs. 50–52). The large positive entropy of activation is probably due to the formation of lower symmetry structures in the transition state and the release of anions and solvent molecules. The

inversion rate obtained here is nearly equivalent to that of [Pd$_3$**L**Cl$_6$] with a loosely-twisted conformation ($k = 14\,\text{s}^{-1}$ at 292 K in CD$_2$Cl$_2$/DMSO-$d_6$)[33], although the ligands and conditions are different. The relatively fast inversion rate of the loosely-twisted complexes is consistent with the argument, discussed earlier, that the helicity inversion does not require configurational inversion of the amine nitrogen atoms.

In contrast, no chemical exchange signals between H$_\text{o}$ and H$_\text{v}$ were observed in the EXSY spectra of **1**$_\text{tight}$ in acetone-$d_6$ at 300 K (Fig. 4d–f). This suggests that the inversion rate of **1**$_\text{tight}$ is too slow to be evaluated by EXSY. Compared to **1**$_\text{loose}$, the helicity inversion of **1**$_\text{tight}$ requires configurational inversion of all amine nitrogen atoms, as described above, which slows the inversion rate. The triflate incorporated into the interior of the macrocycle may also contribute to stabilising the absolute configuration of the inward amine nitrogen atoms via hydrogen bonding. Therefore, we next attempted to synthesise enantio-enriched **1**$_\text{tight}$ using a chiral auxiliary and estimated its racemisation rate as the helicity inversion rate.

## Asymmetric synthesis of 1$_\text{tight}$

Asymmetric synthesis of enantio-enriched **1**$_\text{tight}$ was investigated using chiral sulfoxides as additives (Fig. 5). Among several chiral sulfoxides examined, ($R/S$)-mesityl methyl sulfoxide (($R/S$)-**3**) was the best in

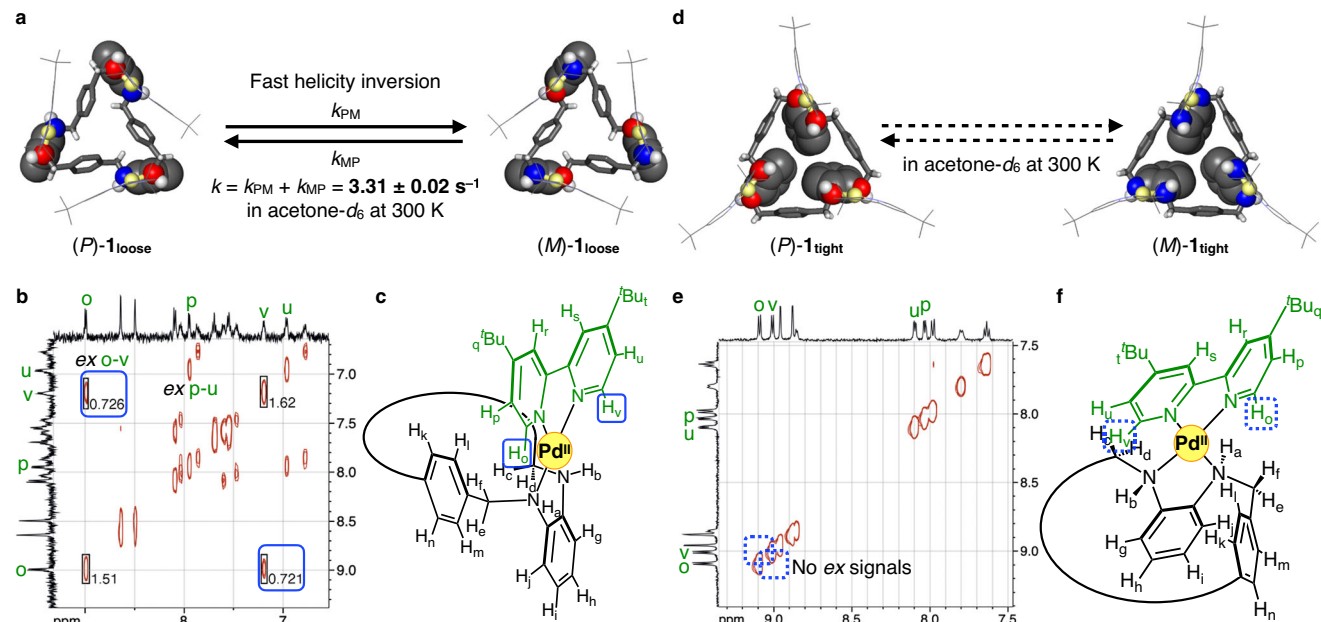

**Fig. 4 | Estimation of the helicity inversion rate. a** Scheme of the helicity inversion between (*P*)- and (*M*)-**1_loose**. **b** EXSY spectrum (500 MHz, acetone-$d_6$, 300 K, mixing time = 0.3 s, 0.19 mM) of (*P/M*)-**1_loose**. The chemical exchange signals between $H_o$ and $H_v$ (*ex o-v*), flamed in blue, were used to estimate the inversion rate. **c** Partial structural formula of **1_loose**. **d** Scheme of the helicity inversion between (*P*)- and (*M*)-**1_tight**. **e** EXSY spectrum (500 MHz, acetone-$d_6$, 300 K, mixing time = 0.3 s) of (*P*)- and (*M*)-**1_tight**. No chemical exchange signals between $H_o$ and $H_v$ were observed, as shown in the blue dotted boxes. **f** Partial structural formula of **1_tight**.

terms of product yield and optical purity (Fig. 5c, d and Supplementary Fig. 96). Specifically, 3.2 equiv. of [Pd($^t$Bu$_2$bpy)(OH$_2$)$_2$](OTf)$_2$ was mixed with an excess of (*S*)-**3** and reacted with **L** in CH$_2$Cl$_2$ at −70 °C for 4 h (Fig. 5a). After washing (*S*)-**3**, enantio-enriched **1_tight** was obtained in 15% yield. The enantiomeric excess of the product was determined to be 25% ee by $^1$H NMR in CD$_2$Cl$_2$ containing Δ-TRISPHAT tetra-butylammonium salt (Δ-**4**) as a chiral shift reagent (Fig. 5e, f and Supplementary Fig. 90). Δ-**4** had no effect on the enantiomeric ratios of the products, as evidenced by time-course $^1$H NMR analysis of racemic and enantio-enriched **1_tight** (Supplementary Figs. 114–117). The product was pure enough for further analysis and was not recrystallised to prevent changes in enantiomeric excess. When analysed by circular dichroism (CD) spectroscopy, this product exhibited a negative Cotton effect at 325 nm in CH$_2$Cl$_2$ (Fig. 5b (red line)). Using enantiomer (*R*)-**3**, the other enantio-enriched **1_tight** with opposite chirality was synthesised in the same way, and the product showed a positive Cotton effect at 325 nm in the CD spectrum (Fig. 5b (blue line)). The mirror image of the CD spectra indicated that the asymmetric synthesis was successfully achieved by chiral **3**.

Time dependent-DFT calculations [M06-D3/def2svp for Pd, 6-31G(d) for other atoms] were then performed to determine the absolute structures of both enantio-enriched **1_tight**. The calculated CD spectrum of the optimised (*P*)-**1_tight** qualitatively reproduced the experimental spectrum of the enantio-enriched **1_tight** synthesised with (*R*)-**3** (Supplementary Fig. 121). For example, positive Cotton effects in the low energy region were found in both the experimental and cal-culated spectra. Similar results were obtained in the calculations of CD spectra with other functionals or basis sets (Supplementary Figs. 122–125). These support that the (*P*)- and (*M*)-enantio-enriched **1_tight** were synthesised with (*R*)- and (*S*)-**3**, respectively.

## Helicity inversion versus twist loosening observed in 1_tight (Fig. 6)

The helicity inversion rate was evaluated using (*M*)-enantio-enriched **1_tight**. An acetone-$d_6$ solution of the (*M*)-enantio-enriched **1_tight** (*P:M* = 37:63) was allowed to stand at 293 K for 3 days (Fig. 5g). After the solvent was removed, the enantiomeric ratio was examined using Δ-**4**

in CD$_2$Cl$_2$, and its enantiomeric ratio (*P:M* = 38:62) was almost the same as that of the starting material. This indicates that either the helicity inversion is too slow to be detected or that the inversion of **1_tight** does not occur under this condition (Fig. 5f, Supplementary Sec-tions 7.1–7.2). Similarly, the inversion rate in CD$_2$Cl$_2$ was also examined at 293 K, but no inversion was observed in 10 days (Supplementary Figs. 112 and 113). Besides, **1_tight** gradually isomerised to **1_loose** as described above, and the isomerisation to **1_loose** was observed during these analyses. These results suggest that the rate of isomerisation from **1_tight** to **1_loose** (twist loosening) ($5.7 \times 10^{-6}$ s$^{-1}$) is faster than that between (*P*)- and (*M*)-**1_tight** (helicity inversion). The faster isomerisation from **1_tight** to **1_loose** can be explained from the number of amine nitrogen atoms whose absolute configuration inverts. That is, in the case of (*all-R*) or (*all-S*) → (*alt-R/S*) (twist loosening), only three of the six amine portions need to be inverted, but in the case of (*all-R*) ⇄ (*all-S*) (helicity inversion), all six amine portions must be inverted. Since configurational inversion of the amine moieties involves dissociation of the N–Pd or N–H bonds, the number of nitrogen inversion sites may affect the isomerisation rate. This consideration is also applied to understanding that the helicity inversion of **1_loose** (1.38 s$^{-1}$ at 293 K, estimated by the Eyring plot, Supplementary Fig. 52) is much faster than that of **1_tight**. This is because the helicity inversion of **1_loose** does not require the configurational changes of amine nitrogen atoms ((*alt-R/S*) ⇄ (*alt-R/S*)). On the other hand, the inversion of the *all-R* or *all-S* configuration in **1_tight** probably needs to occur in a stepwise manner via the intermediary *alt-R/S* configuration, but the intermediate corre-sponding to thermodynamically stable **1_loose** is no longer isomerised to the *all-S* or *all-R* configuration, respectively (Fig. 7).

Finally, **1_tight** was found to be more stabilised when the solvent was substituted from acetone-$d_6$ to CD$_2$Cl$_2$. The stability of (*M*)-enan-tio-enriched **1_tight** was examined by $^1$H NMR spectroscopy and it was found that in acetone-$d_6$ twist loosening and degradation began within 1 day at 293 K, whereas in CD$_2$Cl$_2$ such changes were significantly slower even after 6 days (Supplementary Figs. 109 and 112). Of note, the addition of Δ-**4** in CD$_2$Cl$_2$ also markedly inhibited the tight-to-loose isomerisation. $^1$H NMR spectra of (*M*)-enantio-enriched **1_tight** in the presence of Δ-**4** (17 equiv.) showed that most of **1_tight** remained

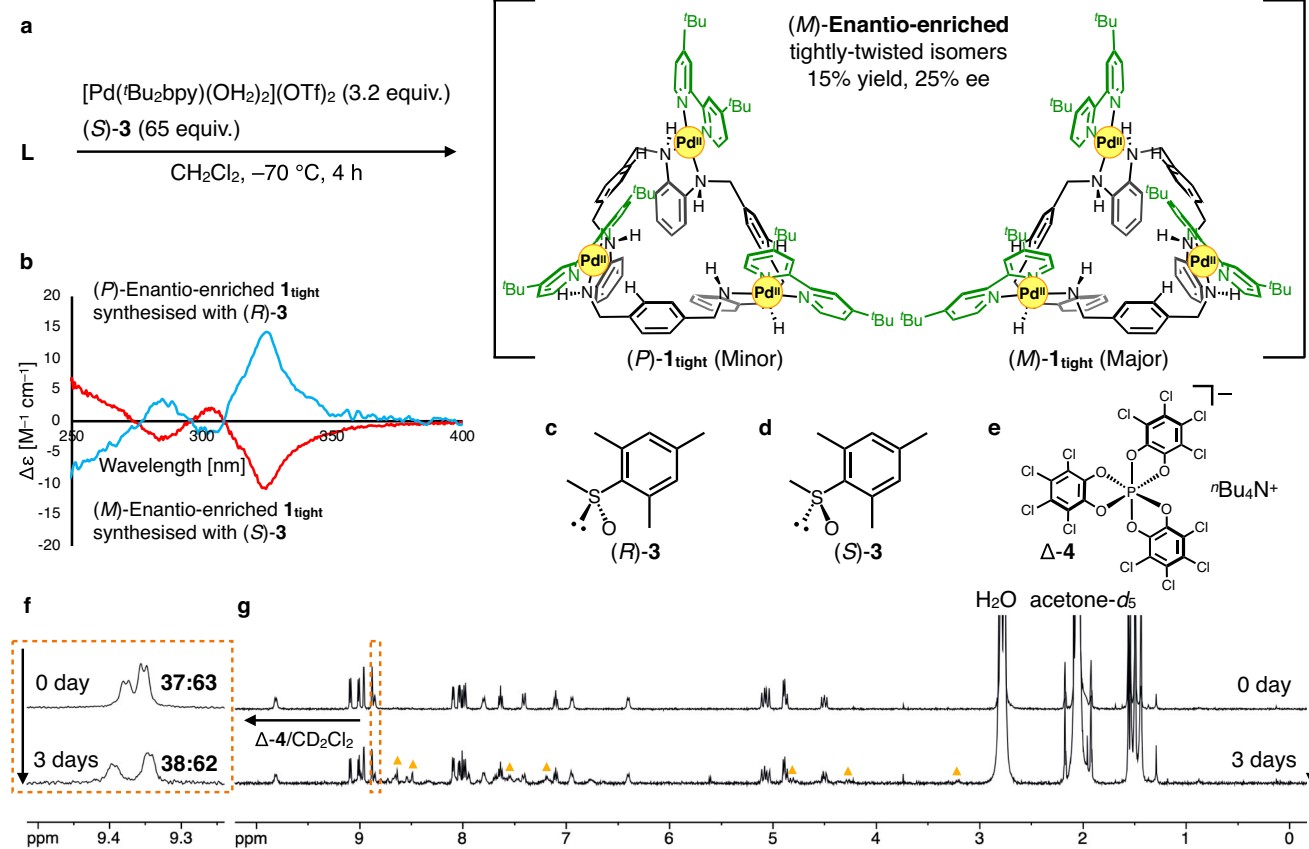

**Fig. 5 | Asymmetric synthesis of $1_{tight}$.** **a** Synthesis of (*M*)-enantio-enriched $1_{tight}$. **b** CD spectra ($CH_2Cl_2$, 293 K, *l* = 1.0 cm) of (*P*)- and (*M*)-enantio-enriched $1_{tight}$ (blue and red lines, respectively). Structural formula of (*R*)-**3** **c**, (*S*)-**3** **d** and Δ-TRISPHAT tetrabutylammonium salt (Δ-**4**) **e**. **f** Partial $^1H$ NMR spectra (500 MHz, $CD_2Cl_2$, 300 K) of the (*M*)-enantio-enriched $1_{tight}$ with Δ-**4**. The upper spectrum (day 0) was recorded by dissolving the as-synthesised product in $CD_2Cl_2$ containing Δ-**4**. The lower spectrum (day 3) was obtained by dissolving the product in acetone-$d_6$, allowing it to stand at 293 K for 3 days, evaporating the solution at room temperature and then redissolving in $CD_2Cl_2$ containing Δ-**4**. Day 0 and day 3 enantiomer excesses were evaluated by the integral of the signals after deconvolution. **g** $^1H$ NMR analysis (500 MHz, acetone-$d_6$, 300 K) of the (*M*)-enantio-enriched $1_{tight}$ without Δ-**4** dissolved in acetone-$d_6$ after 0 day (upper) and 3 days (lower). Signals of $1_{loose}$ are indicated by orange triangles.

without significant isomerisation or degradation at 293 K in $CD_2Cl_2$ even after 14 days (Supplementary Fig. 116). One possibility is that the Δ-TRISPHAT anion associates with the cationic $1_{tight}$ with a similar symmetry to increase the stability, which was supported by the shift and splitting of the $^1H$ NMR and $^{19}F$ NMR signals of the encapsulated triflate (Supplementary Figs. 89 and 115). Thus, a very slow or no inversion can be switched to a faster inversion by isomerisation from a tight to a loose state, and the switching speed can be adjusted by additives or solvents.

## Discussion

In this study, we have succeeded in selectively synthesising two twisted isomers of trinuclear $Pd^{II}$-macrocycles with markedly different rates of helicity inversion. In the tightly-twisted isomer $1_{tight}$, the three *ortho*-phenylenediamine moieties were folded inside the macrocyclic skeleton and the absolute configurations of the amine nitrogen atoms were *all-R* or *all-S*, while in the loosely-twisted isomers $1_{loose}$, the three *ortho*-phenylenediamine moieties were folded outside the skeleton and the absolute configurations of the amine nitrogen atoms were *alt-R/S*. In stark contrast to $1_{tight}$, which shows a very slow or no inversion, $1_{loose}$ exhibits fast helicity inversion (1.38 s$^{-1}$ at 293 K in acetone-$d_6$). Moreover, the inversion kinetics can be controlled by isomerisation in a range from $1_{tight}$, where no inversion is detected, to $1_{loose}$, where inversion is fast. Our approach to the control of helicity inversion motion by the twisted isomers resulting from the configurational locking with metal ions is quite different from conventional approaches that require chemical substitutions or additives to control twisting motions. The new strategy of controlling molecular motions by the mode of twisting with coordinating atoms of different configuration is expected to be applicable to a variety of systems and can be expanded to the design of more sophisticated molecular machines.

## Methods

### Synthesis of $1_{tight}$

A $CH_2Cl_2$ solution (1.0 mL) of **L** (10.0 mg, 15.9 μmol, 1.0 equiv.) was mixed with a $CH_2Cl_2$ solution (2.0 mL) of [Pd($^tBu_2$bpy)(OH$_2$)$_2$] (OTf)$_2$·(H$_2$O)$_2$ (38.2 mg, 51.3 μmol, 3.2 equiv.), and then stirred at room temperature for 4 h. During the reaction, the colour of the solution was changed from pale yellow to purple in a few minutes. The reaction mixture was filtered to remove the precipitate and the filtrate was evaporated. The resulting solid was washed with $CHCl_3$ and the residue was dried up under reduced pressure. The solid was recrystallised from $CH_2Cl_2$ by vapour diffusion of Et$_2$O. The obtained plate crystals were washed with a small amount of $CHCl_3$ and dried up under reduced pressure to afford tightly-twisted $1_{tight}$, [Pd$_3$**L**($^tBu_2$bpy)$_3$] (OTf)$_6$·(H$_2$O)$_{4.9}$·(Et$_2$O)$_{0.15}$, (14.3 mg, 5.19 μmol, 33% yield) as a colourless solid.

Mp: > 257 °C (decomp.). $^1H$ NMR (500 MHz, acetone-$d_6$, 300 K): δ 9.81 (d, *J* = 7.0 Hz, 3H), 9.09 (d, *J* = 6.0 Hz, 3H), 9.00 (d, *J* = 6.0 Hz, 3H), 8.95 (d, *J* = 1.5 Hz, 3H), 8.88 (d, *J* = 2.0 Hz, 3H), 8.85 (d, *J* = 3.5 Hz, 3H), 8.10 (dd, *J* = 5.0, 1.0 Hz, 3H), 8.03 (dd, *J* = 6.0, 1.5 Hz, 3H), 7.98 (d, *J* = 8.5 Hz, 3H), 7.80 (d, *J* = 5.5 Hz, 3H), 7.64 (t, *J* = 8.0 Hz, 3H), 7.41 (d,

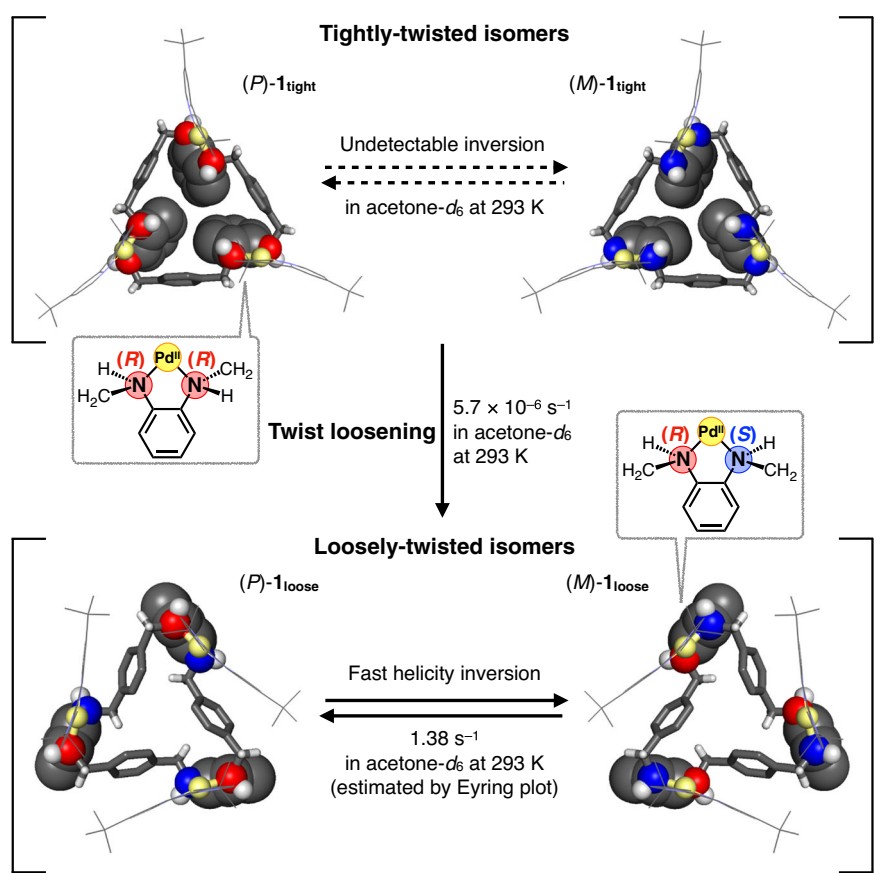

**Fig. 6 | Helicity inversion versus twist loosening of 1$_{tight}$.** Undetectable helicity inversion between (*P*)- and (*M*)-**1**$_{tight}$, twist loosening from **1**$_{tight}$ to **1**$_{loose}$ and fast helicity inversion between (*P*)- and (*M*)-**1**$_{loose}$.

$J$ = 10.5 Hz, 3H), 7.10 (t, $J$ = 7.5 Hz, 3H), 6.94 (d, $J$ = 7.0 Hz, 3H), 6.40 (d, $J$ = 7.0 Hz, 3H), 5.09 (d, $J$ = 14.5 Hz, 3H), 5.05 (d, $J$ = 15.0 Hz, 3H), 4.89 (d, $J$ = 7.5 Hz, 3H), 4.87 (d, $J$ = 13.5 Hz, 3H), 4.50 (dd, $J$ = 14.0, 11.5 Hz, 3H), 1.56 (s, 27H), 1.50 (s, 27H). $^{13}$C NMR (126 MHz, acetone-$d_6$, 301 K): $\delta$ 168.9, 168.7, 157.8, 157.3, 152.2, 150.4, 142.0, 141.8, 134.5, 134.2, 133.4, 133.1, 132.1, 131.9, 131.2, 129.2, 128.6, 126.9, 126.6, 126.5, 123.3, 123.2, 122.9, 120.8, 118.2, 63.0, 61.3, 37.1, 37.0. The $^{13}$C signals of *tert*-butyl groups were overlapped with those of acetone-$d_6$ and could not be identified. The TfO anion has four $^{13}$C signals but only three signals were observed due to the low S/N ratio. $^{19}$F NMR (471 MHz, acetone-$d_6$, 300 K): $\delta$ −75.4, −76.1. IR (ATR, cm$^{-1}$): 3144 (br), 2967, 1618, 1417, 1248, 1155, 1028, 810, 636. UV-vis (CH$_2$Cl$_2$, 293 K, 87.3 μM): $\lambda_{max}$ (nm) ($\varepsilon$ (M$^{-1}$ cm$^{-1}$)) = 309.8 (3.98 × 10$^4$). HRMS (ESI-TOF): $m/z$ = 1100.2498 as [Pd$_3$(H$_{-2}$**L**)($^t$Bu$_2$bpy)$_3$](OTf)$_3$]$^+$ (calcd 1100.2462). Anal. Calcd for C$_{102.6}$H$_{125.3}$F$_{18}$N$_{12}$O$_{23.05}$Pd$_3$S$_6$ {[Pd$_3$**L**($^t$Bu$_2$bpy)$_3$](OTf)$_6$·(H$_2$O)$_{4.9}$·(Et$_2$O)$_{0.15}$}: C 44.82, H 4.58, N 6.09; found: C 44.81, H 4.58, N 6.09.

Crystal data for tightly-twisted Pd$_3$**L**($^t$Bu$_2$bpy)$_3$·(OTf)$_{5.08}$·(H$_2$O)$_{6.95}$·(CH$_2$Cl$_2$)$_{1.27}$ (missing triflates were not observed due to severe disorder): C$_{102.36}$H$_{116.55}$Cl$_{2.55}$F$_{15.25}$N$_{12}$O$_{22.20}$Pd$_3$S$_{5.08}$, $F_w$ = 2732.41, crystal dimensions 0.131 × 0.081 × 0.031 mm$^3$, trigonal, space group *R*-3, $a$ = 23.0889(2), $c$ = 41.6772(6) Å, $V$ = 19241.3(4) Å$^3$, $Z$ = 6, $\rho_{calcd}$ = 1.415 g cm$^{-3}$, $\mu$ = 53.78 cm$^{-1}$, $T$ = 93 K, $\lambda$(CuK$\alpha$) = 1.54187 Å, $2\theta_{max}$ = 144.478°, 39889/8311 reflections collected/unique ($R_{int}$ = 0.0578), $R_1$ = 0.0845 ($I$ > 2$\sigma(I)$), $wR_2$ = 0.2630 (for all data), GOF = 1.113, largest diff. peak and hole 1.414/−1.123 eÅ$^{-3}$. CCDC deposit number 2190130.

### Synthesis of 1$_{loose}$

A CHCl$_3$ solution (2.2 mL) of **L** (13.2 mg, 20.9 μmol, 1.0 equiv.) was mixed with a CHCl$_3$ solution (4.7 mL) of [Pd($^t$Bu$_2$bpy)(OH$_2$)$_2$]

(OTf)$_2$·(H$_2$O)$_2$ (25.2 mg, 33.8 μmol, 1.6 equiv.), and then stirred at room temperature for 3 h. During the reaction, a pink solid was precipitated. The resulting precipitate was collected by filtration and washed with CHCl$_3$ to obtain a dark pink solid whose main component was dinuclear metallocycle **2** (21.3 mg, 10.7 μmol, 64%), which was then suspended in CH$_2$Cl$_2$ (10 mL). To the suspension was added a CH$_2$Cl$_2$ solution (5 mL) of [Pd($^t$Bu$_2$bpy)(OH$_2$)$_2$](OTf)$_2$·(H$_2$O)$_2$ (9.7 mg, 13.0 μmol, 1.2 equiv. to the dinuclear metallocycle). The reaction mixture was stirred at room temperature for 3 h, and then heated at reflux for 1.5 h. During heating, a colourless solid was precipitated. The resulting precipitate was collected by filtration and dried under reduced pressure. This solid was recrystallised from acetone by vapour diffusion of Et$_2$O to afford loosely-twisted **1**$_{loose}$, [Pd$_3$**L**($^t$Bu$_2$bpy)$_3$](OTf)$_6$·(H$_2$O)$_4$, (14.2 mg, 5.21 μmol, 31% in total) as colourless plate crystals.

Mp: > 272 °C (decomp.). $^1$H NMR (500 MHz, acetone-$d_6$, 300 K): $\delta$ 8.99 (d, $J$ = 6.0 Hz, 3H), 8.69 (s, 3H), 8.62 (d, $J$ = 1.5 Hz, 3H), 8.47 (d, $J$ = 1.5 Hz, 3H), 8.10 (d, $J$ = 8.0 Hz, 3H), 8.00 (brs, 3H), 7.91 (dd, $J$ = 5.5, 1.0 Hz, 3H), 7.82 (brs, 3H), 7.68 (t, $J$ = 7.5 Hz, 3H), 7.62 (brs, 3H), 7.55 (m, 6H), 7.47 (brs, 3H), 7.19 (d, $J$ = 5.0 Hz, 3H), 6.94 (d, $J$ = 5.0 Hz, 3H), 6.75 (brs, 3H), 4.78 (d, $J$ = 13.0 Hz, 3H), 4.23 (d, $J$ = 13.0 Hz, 3H), 3.21 (d, $J$ = 11.5 Hz, 3H), 1.88 (brs, 3H), 1.54 (s, 27H), 1.43 (s, 27H). $^{13}$C NMR (126 MHz, acetone-$d_6$, 300 K): $\delta$ 168.6, 166.9, 158.3, 156.4, 151.4, 150.0, 145.6, 141.7, 135.1, 135.1, 134.8, 134.0, 133.8, 133.6, 131.3, 131.3, 127.3, 125.8, 125.7, 123.8, 123.2, 122.8, 120.7, 118.1, 63.1, 60.4, 36.9, 36.7, 30.8, 30.2. $^1$H NMR signals of *para*-phenylene moieties were not fully assigned at 300 K because the rotation of the *para*-phenylene moieties was so fast that extra cross signals derived from chemical exchange processes appeared in the ROESY spectrum and disturbed the assignment of the signals. So, the $^1$H NMR signals of *para*-phenylene moieties were assigned by 2D $^1$H–$^1$H COSY and ROESY NMR analyses at

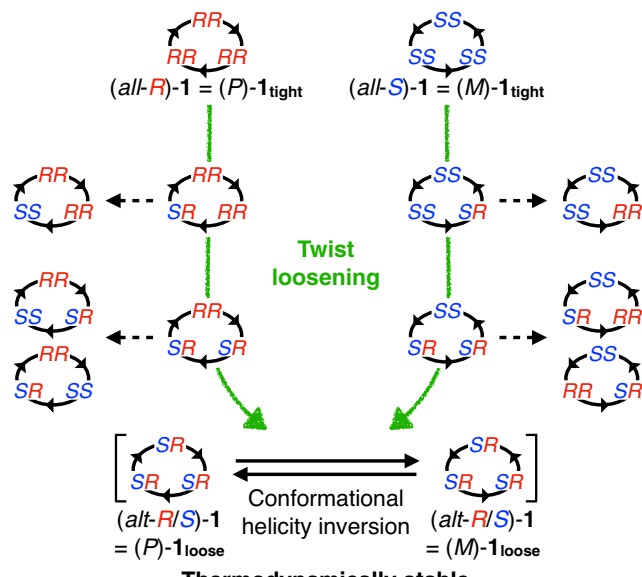

**Fig. 7 | Possible isomers and their isomerisation pathways of 1.** All possible diastereomers and their isomerisation pathways for **1** are shown in this scheme. The black circles indicate the helically twisted macrocyclic structures of **1**. The notions *RR*, *SS* and *SR* above the black circles indicate pairs of the absolute configurations of the amine nitrogen atoms of the *ortho*-phenylenediamine moiety observed in the crystal strictures. The isomerisation pathways to the *RS* configuration are excluded in this scheme, as a pair of the amine nitrogen atoms with the *RS* configurations, with the two amine protons pointing towards the inside of the macrocycle, was not observed experimentally. The green arrows indicate the direct isomerisation pathways from **1**$_{\text{tight}}$ to **1**$_{\text{loose}}$, and the dotted arrows indicate other possible isomerisation pathways.

270 K where the rotation of *para*-phenylene moieties was slow enough. IR (ATR, cm$^{-1}$): 3459 (br), 2975, 1620, 1417, 1240, 1156, 1027, 844. UV-vis (CH$_3$CN, 293 K, 82.3 μM): $\lambda_{\text{max}}$ (nm) ($\varepsilon$ (M$^{-1}$ cm$^{-1}$)) = 311 (3.69 × 10$^4$). HRMS (ESI-TOF): $m/z$ = 1100.2410 as [Pd$_3$(H$_{-1}$**L**)($^t$Bu$_2$bpy)$_3$](OTf)$_4$]$^+$ (calcd 1100.2462). Anal. Calcd for C$_{102}$H$_{122}$F$_{18}$N$_{12}$O$_{22}$Pd$_3$S$_6$ {[Pd$_3$**L**($^t$Bu$_2$bpy)$_3$](OTf)$_6$·(H$_2$O)$_4$}: C 45.01, H 4.52, N 6.18; found: C 45.04, H 4.52, N 6.17.

Crystal data for loosely-twisted Pd$_3$**L**($^t$Bu$_2$bpy)$_3$·(OTf)$_6$·(C$_3$H$_6$O)$_{0.375}$·(H$_2$O)$_{0.5}$: C$_{103.12}$H$_{116.25}$F$_{18}$N$_{12}$O$_{18.88}$Pd$_3$S$_6$, $F_w$ = 2679.38, crystal dimensions 0.221 × 0.150 × 0.087 mm$^3$, trigonal, space group *P*-3*c*1, $a$ = 23.8678(5), $c$ = 29.5387(5) Å, $V$ = 14572.9(7) Å$^3$, $Z$ = 4, $\rho_{\text{calcd}}$ = 1.221 g cm$^{-3}$, $\mu$ = 44.34 cm$^{-1}$, $T$ = 93.15 K, $\lambda$(CuK$\alpha$) = 1.54178 Å, 2$\theta_{\text{max}}$ = 134.130°, 30304/8396 reflections collected/unique ($R_{\text{int}}$ = 0.0431), $R_1$ = 0.1399 ($I > 2\sigma(I)$), $wR_2$ = 0.3524 (for all data), GOF = 1.473, largest diff. peak and hole 6.128/−2.917 eÅ$^{-3}$. CCDC deposit number 2190129.

**Asymmetric synthesis of (*M*)-enantio-enriched 1$_{\text{tight}}$ from (*S*)-3**
A CH$_2$Cl$_2$ solution (0.5 mL) of (*S*)-**3** (96% ee, 93.5 mg, 513 μmol, 65 equiv.) was added to a CH$_2$Cl$_2$ solution (1.0 mL) of [Pd($^t$Bu$_2$bpy)(OH$_2$)$_2$] (OTf)$_2$·(H$_2$O)$_2$ (19.0 mg, 25.5 μmol, 3.2 equiv.), and this reaction mixture was then cooled to −70 °C. To this solution was added a CH$_2$Cl$_2$ solution (1.0 mL) of **L** (5.0 mg, 7.93 μmol, 1.0 equiv.) at −70 °C. This reaction mixture was stirred at −70 °C for 4 h. After bringing the reaction mixture to room temperature, Et$_2$O was added and the reaction mixture was filtered. The filtrate was evaporated under reduced pressure to afford (*S*)-mesityl methyl sulfoxide (87.9 mg, 482 μmol, 94% recovery yield, 95% ee). The precipitate was dissolved in CH$_2$Cl$_2$ and filtered to remove insoluble residue. The filtrate was evaporated and dried in vacuo. After the resulting solid was washed with CHCl$_3$, (*M*)-enantio-enriched **1**$_{\text{tight}}$ was obtained as a pale yellow solid (3.2 mg, 1.19 μmol,

15% yield, 25% ee). The enantiomeric excess was estimated using Δ-**4** as a chiral shift reagent.

## Data availability
All data obtained and analysed are available in this Article and the Supplementary information. Crystallographic data for the structures reported in this Article have been deposited at the Cambridge Crystallographic Data Centre, under deposition numbers CCDC 2190129 (**1**$_{\text{loose}}$), 2190130 (**1**$_{\text{tight}}$), 2190131 ([Pd($^t$Bu$_2$bpy)(OH$_2$)$_2$](OTf)$_2$)), 2190132 ([Pt$_2$**L**($^t$Bu$_2$bpy)$_2$](OTf)$_4$), 2190133 (1,2,5,6-di-*O*-iso-propylidene-α-D-glucofuranosyl (*S*)-methanesulfinate) and 2190134 ((*R*)-*o*-anisyl methyl sulfoxide). Copies of the data can be obtained free of charge via https://www.ccdc.cam.ac.uk/structures/. All other data are available from the corresponding author upon request.

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

## Acknowledgements

This research was supported by the JSPS KAKENHI, grant number JP16H06509 (Coordination Asymmetry) to M.S. and by the Asahi Glass Foundation to S.T. The computational studies were performed using Research Centre for Computational Science, Okazaki, Japan (Project: 21-IMS-C208, 22-IMS-C175). T.N. thanks Global Science Graduate Course (GSGC).

## Author contributions

T.N. performed the experimental work. T.N. and M.E. contributed to the computational studies. T.N., S.T. and M.S. designed the project, analysed the data and prepared the manuscript.

## Competing interests

The authors declare no competing interests.
