## [Peer Review File · Nature Communications]

Selective synthesis of tightly- and loosely-twisted metallomacrocycle isomers towards precise control of helicity inversion motionReviewers' Comments:

Reviewer #1:

Remarks to the Author:

This manuscript by Shionoya and coworkers reports a controlled syntheses and molecular twisting motion by rewinding helicity inversion of an interesting pair of topological-isomeric Palladium-organic macrocycles. Selectively syntheses of the two isomers, i.e. 1-tight and 1-loose, with the same organic and metal components are somehow by serendipity but it is often the case for amazing chemistry to come. More and more studies in this field suggest that the inherent thermodynamic and kinetic parameters for a given self-assembly processes are hard to predict. Interestingly, EXSY NMR studies showed that the loosely-twisted isomers smoothly rewind between (P)- and (M) helicity while the helicity inversion of the tightly-twisted isomers is undetectable before it transforms to the loosely-twisted isomers. This remarkable difference between these two isomers have been explained by the presence or absence of an absolute configuration inversion of the nitrogen atoms on the macrocyclic ligand. Effects of solvents, counter-ions and chiral auxiliary ligands have also been revealed for the kinetics of the molecular twisting. Molecular-level understanding on the control of twisting motions is an essential step before achieving more sophisticated molecular machines. I believe this study is of potential interests to a broad readership and thus I recommend acceptance of this nice work on Nature Commun after some minor changes.

Minor points noticed:

- (1) The mixed use on the term of "isomerization" and "helicity inversion" are some-how confusing. I suggest to be more specific to make distinction between the (all-R)/(all-S) \rightarrow (alt-R/S) and (all-R) \rightleftharpoons (all-S) "isomerization".
- (2) Molecular formula for the 1-tight complex in the crystal data {Pd3L(tBu2bpy)3·(OTf)5.08·(H2O)6.95·(CH2Cl2)1.27} need to be corrected.
- (3) Analogous Pt-complexes were reported in the ESI. It will be very useful if the authors add more comparison on the self-assembly, transformation and helicity inversion, between the Pd and Pt-based complexes.
- (4) C-H...Pd anagostic interactions are known to be sensitive to temperature and metal-centers. Maybe VT-NMR can further confirm their existence in solution.

Reviewer #2:

Remarks to the Author:

In this manuscript, Tashiro, Shionoya, and co-workers report on a new pair of isomeric twisted macrocycles, 1_tight and 1_loose, prepared through coordination of a macrocyclic ligand (L) to Pd(II). 1_tight is obtained as a kinetic product and does not undergo inversion, whereas 1_loose does. The complexes thus represent an unusual example of interconverting twisted macrocycles with very different conformational dynamics. The essential difference appears to be the configurations of the N atoms coordinated to Pd. In the "loose" isomer the configurations alternate and thus do not need to invert on helical inversion, whereas in the "tight" isomer the N's are homochiral and must invert along with the helix.

I am quite impressed with how thorough this work is, with extensive crystallographic, NMR, and computational experiments. The conclusions are very well supported by the results, which have been fully documented in the SI. My comments below relate only to the discussion and presentation. I believe this work can be published after minor revisions.

My main comment is about the distinction between the "tight" and "loose" twisting of the macrocycles. This is done based on the orientation of the phenylenediamine moieties but I don't really see how this makes one macrocycle more tightly twisted than the other. Tight vs loose twisting implies a difference in helical pitch, which I don't think we have in these macrocycles.

The idea that a tightly twisted conformer undergoes slower inversion is appealing, but what's happening here is related to the stabilization of the amine configurations by the metal. The metals lock the configurations of the amines, and that affects the folding of the rest of the structure. To me that idea is more interesting and probably more applicable to other systems. Focusing on the twist risks hiding this.

I would also suggest including the structure of the previously reported [Pd3LCI6] as the manuscript makes several comparisons to its geometry. Hydrogens should also be removed from geometries throughout the manuscript as they make the figures difficult to interpret.

This statement (line 138) is a bit confusing: "In contrast, 1_tight was less stable and slowly isomerised to 1_loose in acetone-d6 to produce non-assignable by-products." I think that what is meant is that 1_tight undergoes isomerization to 1_loose while also producing non-assignable by-products?

Reviewer #3:

Remarks to the Author:

In this manuscript, the authors report the two types of trinuclear Pd complexes having a macrocyclic N6 ligand. One is the tightly-twisted complex that exhibits no helicity inversion, and the other is the loosely-twisted complex that exhibits a rapid helicity inversion. The authors carefully characterized the two types of complexes and appropriately analyzed the helix inversion rates by NMR EXSY measurements and time dependent NMR measurements.

The observed isomerism reported in this paper is interesting, but unfortunately, this reviewer did not agree with authors' claim based on the points described below. The authors' claim would be correct and have a great significance if the tight/loose isomerism was ascribed solely to the different twisting modes without stereochemical isomerization at the nitrogen centers and the helicity inversion of the two forms occurs in an independent isomerization pathway, but this is not the case. Therefore, the argument in this paper is not acceptable to be published in Nature Communications.

Detailed explanations

1. The two isomers, tightly- and loosely-twisted isomers, were not actually a pair of "twisted isomers", but nitrogen-centered stereoisomers that have different twisting conformations. In lines 45-47, the authors described that "it has not been realised that such twisting motions are controlled by the mode of twisting", but the reported twisting motions are not controlled by "twisting" but just by the nitrogen-centered isomerism, e.g., RR -> RS -> SS and vice versa.

2. As the authors mentioned, the "helicity inversion" of the tightly-twisted isomer requires inversion of the stereochemistry between the RR and SS configurations of the amine nitrogen atoms for each Pd center. This means that the RR configuration can be inverted into the SS form only via the RS/SR configuration if cleavage/rebound of the Pd-N bonds occurs in a stepwise manner. The loosely-twisted isomer having the alt-R/S configuration is the most probable intermediate during the possible helicity inversion of the tightly-twisted all-R and all-S forms, although this helicity inversion was not observed due to the relative thermodynamic stability of the loosely-twisted form. Therefore, the irreversible transformation to the loosely-twisted form is inevitable during the "helicity inversion" of the tightly-twisted forms, and the helicity inversion pathways of the two isomers cannot be separately discussed.

Reviewer #1's comments and our responses

This manuscript by Shionoya and coworkers reports a controlled syntheses and molecular twisting motion by rewinding helicity inversion of an interesting pair of topological-isomeric Palladium-organic macrocycles. Selectively syntheses of the two isomers, i.e. 1-tight and 1-loose, with the same organic and metal components are somehow by serendipity but it is often the case for amazing chemistry to come. More and more studies in this field suggest that the inherent thermodynamic and kinetic parameters for a given self-assembly processes are hard to predict. Interestingly, EXSY NMR studies showed that the loosely-twisted isomers smoothly rewind between (P)- and (M) helicity while the helicity inversion of the tightly-twisted isomers is undetectable before it transforms to the loosely-twisted isomers. This remarkable difference between these two isomers have been explained by the presence or absence of an absolute configuration inversion of the nitrogen atoms on the macrocyclic ligand. Effects of solvents, counter-ions and chiral auxiliary ligands have also been revealed for the kinetics of the molecular twisting. Molecular-level understanding on the control of twisting motions is an essential step before achieving more sophisticated molecular machines. I believe this study is of potential interests to a broad readership and thus I recommend acceptance of this nice work on Nature Commun after some minor changes.

Minor points noticed:

(1) The mixed use on the term of “isomerization” and “helicity inversion” are some-how confusing. I suggest to be more specific to make distinction between the $(all-R)/(all-S) \rightarrow (alt-R/S)$ and $(all-R) \rightleftharpoons (all-S)$ “isomerization”.

Our responses:

Thank you for your positive feedback on this study and for some important suggestions. We agree with avoiding confusion between the terms “isomerisation” and “helicity inversion” and the textual description. As you point out, both $(all-R)/(all-S) \rightarrow (alt-R/S)$ and $(all-R) \rightleftharpoons (all-S)$ are isomerisation, so it is not appropriate to simply describe $(all-R)/(all-S) \rightarrow (alt-R/S)$ as “isomerisation”. Therefore, this section has been modified to avoid this confusion by distinguishing $(all-R)/(all-S) \rightarrow (alt-R/S)$ (isomerisation from $\mathbf{1}_{tight}$ to $\mathbf{1}_{loose}$) as “twist loosening” and $(all-R) \rightleftharpoons (all-S)$ (isomerisation between (P)- and (M)- $\mathbf{1}_{tight}$) as “helicity inversion”, except where the type of isomerisation is defined.

Correction to Fig. 6 and explanation in text (page 13) (yellow parts: correction related to this point)

Helicity inversion versus isomerisation twist loosening observed in $\mathbf{1}_{tight}$ (Fig. 6). The helicity inversion rate was evaluated using (M)-enantio-enriched $\mathbf{1}_{tight}$. An acetone- d_6 solution of the (M)-enantio-enriched $\mathbf{1}_{tight}$ ($P:M = 37:63$) was allowed to stand at 293 K for 3 days (Fig. 5g). After the solvent was removed, the enantiomeric ratio was examined using $\Delta\text{-4}$ in CD_2Cl_2 , and its enantiomeric ratio ($P:M = 38:62$) was almost the same as that of the starting material. This indicates that either the helicity inversion is too slow to be detected or that the inversion of $\mathbf{1}_{tight}$ does not occur under this condition (Fig. 5f, Supplementary Sections 7.1–7.2). Similarly, the inversion rate in CD_2Cl_2 was also examined at 293 K, but no inversion was observed in 10 days (Supplementary Figs. 112 and 113). Besides, $\mathbf{1}_{tight}$ gradually isomerised to $\mathbf{1}_{loose}$ as described above, and the isomerisation to $\mathbf{1}_{loose}$ was observed during these analyses. These results suggest that the rate of isomerisation from $\mathbf{1}_{tight}$ to $\mathbf{1}_{loose}$ (twist loosening) ($5.7 \times 10^{-6} \text{ s}^{-1}$)

¹) is faster than that between (*P*)- and (*M*)-**1_{tight}** (helicity inversion). The faster isomerisation from **1_{tight}** to **1_{loose}** can be explained from the number of amine nitrogen atoms whose absolute configuration inverts. That is, in the case of (*all-R*) or (*all-S*) → (*alt-R/S*) (twist loosening), only three of the six amine portions need to be inverted, but in the case of (*all-R*) ⇌ (*all-S*) (helicity inversion), all six amine portions must be inverted. Since configurational inversion of the amine moieties involves dissociation of the N–Pd or N–H bonds, the number of nitrogen inversion sites may affect the isomerisation rate. This consideration is also applied to understanding that the helicity inversion of **1_{loose}** (1.38 s⁻¹ at 293 K, estimated by the Eyring plot, Supplementary Fig. 52) is much faster than that of **1_{tight}**. This is because the helicity inversion of **1_{loose}** does not require the configurational changes of amine nitrogen atoms ((*alt-R/S*) ⇌ (*alt-R/S*)). On the other hand, the inversion of the *all-R* or *all-S* configuration in **1_{tight}** probably needs to occur in a stepwise manner via the intermediary *alt-R/S* configuration, but the intermediate corresponding to thermodynamically stable **1_{loose}** is no longer isomerised to the *all-S* or *all-R* configuration, respectively.

Finally, **1_{tight}** was found to be more stabilised when the solvent was substituted from acetone-*d*₆ to CD₂Cl₂. The stability of (*M*)-enatio-enriched **1_{tight}** was examined by ¹H NMR spectroscopy and it was found that in acetone-*d*₆ twist loosening and degradation began within 1 day at 293 K, whereas in CH₂Cl₂ such changes were significantly slower even after 6 days (Supplementary Figs. 109 and 112). Of note, the addition of Δ-4 in CD₂Cl₂ also markedly inhibited the tight-to-loose isomerisation. ¹H NMR spectra of (*M*)-enatio-enriched **1_{tight}** in the presence of Δ-4 (17 equiv.) showed that most of **1_{tight}** remained without significant isomerisation or degradation at 293 K in CD₂Cl₂ even after 14 days (Supplementary Fig. 116). One possibility is that the Δ-TRISPHAT anion associates with the cationic **1_{tight}** with a similar symmetry to increase the stability, which was supported by the shift and splitting of the ¹H NMR and ¹⁹F NMR signals of the encapsulated triflate (Supplementary Figs. 89 and 115). Thus, a very slow or no inversion can be switched to a faster inversion by isomerisation from a tight to a loose state, and the switching speed can be adjusted by additives or solvents.

Fig. 6| Helicity inversion versus twist loosening of 1_{tight}. Undetectable helicity inversion between (P)- and (M)-1_{tight}, twist loosening from 1_{tight} to 1_{loose}, and fast helicity inversion between (P)- and (M)-1_{loose}.

(2) Molecular formula for the 1-tight complex in the crystal data {Pd3L(tBu2bpy)3·(OTf)5.08·(H2O)6.95·(CH2Cl2)1.27} need to be corrected.

Our responses:

Thank you for your important remark. The triflate anions appear to be missing in the molecular formula, but are simply not observed due to severe disorder. To clarify this point, we have added a brief explanation as follows.

Revised part in the text (page 18; Methods, synthesis of 1_{tight}; line 336–337)

Revised parts in the Supplementary Information (page S7)

Crystal data for tightly-twisted $\text{Pd}_3\text{L}(\text{Bu}_2\text{bpy})_3 \cdot (\text{OTf})_{5.08} \cdot (\text{H}_2\text{O})_{6.95} \cdot (\text{CH}_2\text{Cl}_2)_{1.27}$ (missing triflates were not observed due to severe disorder): $\text{C}_{102.36}\text{H}_{116.55}\text{Cl}_{2.55}\text{F}_{15.25}\text{N}_{12}\text{O}_{22.20}\text{Pd}_3\text{S}_{5.08}$,

(3) Analogous Pt-complexes were reported in the ESI. It will be very useful if the authors add more comparison on the self-assembly, transformation and helicity inversion, between the Pd and Pd -based complexes.

Our responses:

Thank you for your important suggestion. We are currently attempting to prepare Pt-analogues of $\mathbf{1}_{\text{loose}}$ and $\mathbf{1}_{\text{tight}}$ by similar synthetic methods for other research purposes, but have not yet succeeded in synthesizing the Pt-analogue of $\mathbf{1}_{\text{tight}}$, probably due to the inherent difficulties associated with the kinetic process. Therefore, it is difficult at this time to discuss the comparison between Pd- and Pt-complexes in terms of self-assembly, transformation and helicity inversion due to the synthetic limitations mentioned above. We believe that the current data presented in this manuscript adequately addresses the discussion of the title. On the other hand, since many readers may have the same impression as your point, we have added a brief description of the difficulty of synthesizing Pt-analogues to the Supplementary Information as follows. Thank you again for your important suggestion.

Description newly added to the Supplementary Information (page S34)

We have attempted to synthesise the Pt-analogue of $\mathbf{1}_{\text{loose}}$ and $\mathbf{1}_{\text{tight}}$ in a similar fashion for comparison, but have not yet succeeded in synthesising the Pt-analogue of $\mathbf{1}_{\text{tight}}$, probably due to the inherent difficulties associated with the kinetic process. Therefore, it is difficult at this time to discuss the differences in structure and dynamics of the Pd- and Pt-complexes of both twisted isomers due to synthetic limitations.

(4) C-H \cdots Pd anagostic interactions are known to be sensitive to temperature and metal-centers. Maybe VT-NMR can further confirm their existence in solution.

Our responses:

Thank you for your useful advice. According to this suggestion, we have shown VT-NMR data in the Supplementary Information and confirmed that the C-H \cdots Pd anagostic interactions are preserved even at 300 K in CD_2Cl_2 based on the temperature dependence of the chemical shift.

VT-NMR newly added to the Supplementary Information as Supplementary Fig. 14 (page S14)

Supplementary Fig. 14 VT ^1H NMR spectra of $\mathbf{1}_{\text{tight}}$ measured at (a) 230 K, (b) 245 K, (c) 260 K, (d) 280 K and (e) 300 K (500 MHz, CD_2Cl_2). The downfield shift of one *para*-phenylene signal (H_l) was maintained in the range from 230 to 300 K, suggesting the presence of C-H \cdots Pd anagostic interactions even at 300 K in the solution.

Reviewer #2's comments and our responses

In this manuscript, Tashiro, Shionoya, and co-workers report on a new pair of isomeric twisted macrocycles, **1_{tight}** and **1_{loose}**, prepared through coordination of a macrocyclic ligand (L) to Pd(II). **1_{tight}** is obtained as a kinetic product and does not undergo inversion, whereas **1_{loose}** does. The complexes thus represent an unusual example of interconverting twisted macrocycles with very different conformational dynamics. The essential difference appears to be the configurations of the N atoms coordinated to Pd. In the "loose" isomer the configurations alternate and thus do not need to invert on helical inversion, whereas in the "tight" isomer the N's are homochiral and must invert along with the helix.

I am quite impressed with how thorough this work is, with extensive crystallographic, NMR, and computational experiments. The conclusions are very well supported by the results, which have been fully documented in the SI. My comments below relate only to the discussion and presentation. I believe this work can be published after minor revisions.

My main comment is about the distinction between the "tight" and "loose" twisting of the macrocycles. This is done based on the orientation of the phenylenediamine moieties but I don't really see how this makes one macrocycle more tightly twisted than the other. Tight vs loose twisting implies a difference in helical pitch, which I don't think we have in these macrocycles.

Our responses:

Thank you for your appropriate assessment of this study and some important suggestions. As you have indicated, it is very important to clearly state the mode of twisting used in this manuscript and the definitions of "tight" and "loose". The definition used here differs from the typical definition of a helix based on the difference in helical pitch. This is because the structure of **1** is not a helix by structural definition, but a twist with a *P*- and an *M*-helicity. Thus in this study, the mode of twisting is defined by how much the *ortho*-phenylenediamine moieties are folded inside the macrocycle. The details of this definition are added in the Supplementary Information to clearly show the difference from the typical definition of helices. This definition also includes differences in the absolute configuration of nitrogen atoms locked by metal coordination in both twisted isomers, which is also relevant to the next point.

Description newly added to the Supplementary Information (page S26)

2.4 Definition of the two twisted isomers of **1**

The two twisted isomers of **1** are defined by the degree to which the *ortho*-phenylenediamine moieties are folded inside the macrocycle. The degree of folding is evaluated by the dihedral angle between the plane horizontal to the macrocyclic skeleton and the plane of the *ortho*-phenylenediamine ring. The dihedral angle in **L** before metal coordination is nearly 0°, indicating no twisting. One isomer of the twisted Pd^{II}₃-macrocycles with a dihedral angle of greater than 90° is defined as the tightly-twisted isomer, **1_{tight}**. In contrast, the other isomer with a dihedral angle of less than 90° is therefore defined as the loosely-twisted isomer, **1_{loose}**. As a result of the different twisting modes, the absolute configuration of the amine nitrogen atoms changes through metal coordination to (*all-R* or *all-S*) and (*alt-R/S*) for the tightly- and loosely-twisted isomers, respectively. Note that this definition differs from the typical definition of a

helix, which is based on differences in helical pitch. This is because the structure of **1** is not a helix, but a twist with the *P*- and *M*-helicity.

Supplementary Fig. 33 Illustration supporting the definition of the twisted isomers: (left) **L** is twisted tightly to form (*P*)-**1**_{tight} forming the (*all-R*) configuration with *ortho*-phenylenediamine nitrogen atoms, and (right) **L** is twisted loosely to form (*P*)-**1**_{loose} with the (*alt-R/S*) configuration with *ortho*-phenylenediamine nitrogen atoms.

The idea that a tightly twisted conformer undergoes slower inversion is appealing, but what's happening here is related to the stabilization of the amine configurations by the metal. The metals lock the configurations of the amines, and that affects the folding of the rest of the structure. To me that idea is more interesting and probably more applicable to other systems. Focusing on the twist risks hiding this.

Our responses:

We very much agree with your comments. This concept of fixing the amine configuration by metal coordination was originally included in our strategy, as presented in the Abstract and Results and Discussion sections, but this point was not adequately explained in the Introduction and Conclusion sections. Therefore, we have made the following changes to the Introduction (including Fig. 1) and Conclusion sections as follows to more accurately convey our strategy. In addition, italics were removed to avoid overemphasising the “mode of twisting” throughout the manuscript.

Corrected second and fourth paragraphs in the Introduction section (page 2)

Second paragraph: We intuitively understand that twisting behaviours are highly dependent on the mode of twisting. For instance, a loosely-twisted object will unwind easily, but a very tightly-twisted object may not. Can such movements be reproduced in the nanoscale world? Inspired by biomacromolecules, chemists have synthesised twisted molecules such as helical polymers^{18–20}, helicenes^{21,22} and twisted macrocycles^{23–25}, and in some of these examples, the design and control of their twisting motion are being studied. For instance, the rate of helicity

inversion was controlled by exploiting the kinetic properties of coordination bonds in twisted metal complexes of synthetic peptides²⁶, macrocycles²⁷ and cryptands²⁸. Another role of metal coordination is to dynamically fix the absolute configuration of the coordinating atoms, including the amine nitrogen atoms, which is usually immediately reversed. Thus, selective synthesis of metal complexes with different modes of twisting resulting in different configurations and/or conformation, or "twisted isomers" is a promising strategy to control the inversion motion without changing chemical composition, but such isomers are limited to a few examples^{29–32}. Thus, it is still challenging to selectively synthesise such twisted isomers.

Fourth paragraph: Herein we report the selective synthesis of two twisted isomers of a trinuclear Pd^{II}-macrocycle with a tightly- or loosely-twisted skeleton (Fig. 1). It is particularly important to emphasise that these two isomers have markedly different rates of helicity inversion depending on the mode of twisting with different absolute configurations of the diamine moieties locked by the metal ions. The loosely-twisted isomer exhibited rapid helicity inversion, whereas the helicity inversion in the tightly-twisted isomer was actually undetectable because the inversion process requires absolute configuration inversion of the nitrogen atoms. In other words, the helicity inversion of the twisted macrocycle is configurationally locked inhibited by twisting more tightly. Thus, this result is an excellent example of twisting motion controlled by the mode of twisting of a single chiral molecule with coordinating atoms of different configuration.

Fig. 1 | The concept of molecular helicity inversion controlled by twisting mode due to differences in the absolute configurations of the diamine moieties locked by metal ions.

Colour: Pd yellow, C black, H white, N Purple (non-coordinated), red (*R*-configuration) and blue (*S*-configuration).

Corrected Conclusion section (page 16)

In this study, we have succeeded in selectively synthesising two twisted isomers of trinuclear Pd^{II}-macrocycles with markedly different rates of helicity inversion. In the tightly-twisted isomer **1_{tight}**, the three *ortho*-phenylenediamine moieties were folded inside the macrocyclic skeleton and the absolute configurations of the amine nitrogen atoms were *all-R* or *all-S*, while in the loosely-twisted isomers **1_{loose}**, the three *ortho*-phenylenediamine moieties were folded outside the skeleton and the absolute configurations of the amine nitrogen atoms were *alt-R/S*. In stark contrast to **1_{tight}**, which shows a very slow or no inversion, **1_{loose}** exhibits fast helicity inversion (1.38 s⁻¹ at 293 K in acetone-*d*₆). Moreover, the inversion kinetics can be controlled by isomerisation in a range from **1_{tight}**, where no inversion is detected, to **1_{loose}**, where inversion is fast. Our approach to the control of helicity inversion motion by the twisted isomers resulting from the configurational locking with metal ions is quite different from conventional approaches that require chemical substitutions or additives to control twisting motions. The new strategy of controlling molecular motions by the mode of twisting with coordinating atoms of different configuration is expected to be applicable to a variety of systems and can be expanded to the design of more sophisticated molecular machines.

I would also suggest including the structure of the previously reported [Pd₃LCI₆] as the manuscript makes several comparisons to its geometry. Hydrogens should also be removed from geometries throughout the manuscript as they make the figures difficult to interpret.

Our responses:

Thank you for your important comments. We have followed your suggestion and added the reported crystal structure of [Pd₃LCI₆] to Fig. 3, removed unnecessary hydrogen atoms from the crystal structures of **1_{tight}** and **1_{loose}** and added the following to more clearly show the difference in molecular geometry.

Corrected Fig. 3 (page 9)

Fig. 3| Characterisation of the two twisted isomers of 1. a, Substructural formula of 1_{tight} . b,c, Side and top views of the crystal structure of 1_{tight} with one triflate incorporated via multipoint hydrogen bonding. Hydrogen atoms, except amine and methylene moieties, are omitted for clarity. d, ^1H NMR spectrum (500 MHz, acetone- d_6 , 300 K) of 1_{tight} . e, Substructural formula of 1_{loose} . f,g, Side and top views of the crystal structure of 1_{loose} with one triflate disordered in the inner space. Hydrogen atoms, except amine and methylene moieties, are omitted for clarity. h, ^1H NMR spectrum (500 MHz, acetone- d_6 , 300 K) of 1_{loose} . i,j, Structural formula and the reported crystal structure (top view) of $[\text{Pd}_3\text{LCl}_6]^{33}$.

This statement (line 138) is a bit confusing: "In contrast, 1_{tight} was less stable and slowly isomerised to 1_{loose} in acetone- d_6 to produce non-assignable by-products." I think that what is meant is that 1_{tight} undergoes isomerization to 1_{loose} while also producing non-assignable by-products?

Our responses:

Thank you for your important comment. We have corrected the text as the reviewer suggested.

Corrected fourth paragraph of the Results and Discussion section (page 6, line 145–146)

1_{loose} was stable in acetone-*d*₆ at room temperature for two weeks, as evidenced by ¹H NMR analysis (Supplementary Fig. 32). In contrast, **1**_{tight} was less stable and slowly isomerised to **1**_{loose} in acetone-*d*₆ ~~to~~ while also producing non-assignable by-products.

Reviewer #3's comments and our responses

In this manuscript, the authors report the two types of trinuclear Pd complexes having a macrocyclic N6 ligand. One is the tightly-twisted complex that exhibits no helicity inversion, and the other is the loosely-twisted complex that exhibits a rapid helicity inversion. The authors carefully characterized the two types of complexes and appropriately analyzed the helix inversion rates by NMR EXSY measurements and time dependent NMR measurements.

The observed isomerism reported in this paper is interesting, but unfortunately, this reviewer did not agree with authors' claim based on the points described below. The authors' claim would be correct and have a great significance if the tight/loose isomerism was ascribed solely to the different twisting modes without stereochemical isomerization at the nitrogen centers and the helicity inversion of the two forms occurs in an independent isomerization pathway, but this is not the case. Therefore, the argument in this paper is not acceptable to be published in Nature Communications.

Detailed explanations

1. The two isomers, tightly- and loosely-twisted isomers, were not actually a pair of "twisted isomers", but nitrogen-centered stereoisomers that have different twisting conformations. In lines 45-47, the authors described that "it has not been realised that such twisting motions are controlled by the mode of twisting", but the reported twisting motions are not controlled by "twisting" but just by the nitrogen-centered isomerism, e.g., RR -> RS -> SS and vice versa.

Our responses:

Thank you for your important comments on our manuscript. Indeed, our strategy of controlling molecular motion by different twisting modes involves not only conformational differences but also configurational differences between two twisted isomers as shown in the Abstract and the Results and Discussion sections. More specifically, the conformational differences due to different twisting modes cause the configurational differences of the coordinating amine atoms ((*all-R*)/(*all-S*) or (*alt-R/S*)) locked by metal coordination, and these differences ultimately lead to distinct differences in the rate of inversion motion. This means that engaging in configurational differences in twisted isomers does not undermine the value of our strategy, but rather enhances it based on its utility and novelty. Therefore, we strongly believe that the significance of this strategy is not undermined at all, even if the configurational differences are involved in the different twisting modes.

On the other hand, it was our fault that the explanation of this point was insufficient, and it may mislead readers and reviewers. Therefore, we have modified the Introduction and Conclusion parts and have also added the definition of the twisted isomers used in this study to the Supplementary Information as follows. Thank you for pointing out this important point that will help us to describe our strategy more accurately.

Corrected second and fourth paragraphs in the Introduction section (page 2)

Second paragraph: We intuitively understand that twisting behaviours are highly dependent on the mode of twisting. For instance, a loosely-twisted object will unwind easily, but a very tightly-twisted object may not. Can such movements be reproduced in the nanoscale world? Inspired by biomacromolecules, chemists have synthesised twisted molecules such as helical polymers¹⁸⁻²⁰, helicenes^{21,22} and twisted macrocycles²³⁻²⁵, and in some of these examples, the design and control of their twisting motion are being studied. For instance, the rate of helicity

inversion was controlled by exploiting the kinetic properties of coordination bonds in twisted metal complexes of synthetic peptides²⁶, macrocycles²⁷ and cryptands²⁸. Another role of metal coordination is to dynamically fix the absolute configuration of the coordinating atoms, including the amine nitrogen atoms, which is usually immediately reversed. Thus, selective synthesis of metal complexes with different modes of twisting resulting in different configurations and/or conformation, or "twisted isomers" is a promising strategy to control the inversion motion without changing chemical composition, but such isomers are limited to a few examples^{29–32}. Thus, it is still challenging to selectively synthesise such twisted isomers.

Fourth paragraph: Herein we report the selective synthesis of two twisted isomers of a trinuclear Pd^{II}-macrocycle with a tightly- or loosely-twisted skeleton (Fig. 1). It is particularly important to emphasise that these two isomers have markedly different rates of helicity inversion, depending on the mode of twisting with different absolute configurations of the diamine moieties locked by the metal ions. The loosely-twisted isomer exhibited rapid helicity inversion, whereas the helicity inversion in the tightly-twisted isomer was actually undetectable because the inversion process requires absolute configuration inversion of the nitrogen atoms. In other words, the helicity inversion of the twisted macrocycle is configurationally inhibited locked by twisting more tightly. Thus, this result is an excellent example of twisting motion controlled by the mode of twisting of a single chiral molecule with coordinating atoms of different configurations.

Fig. 1 | The concept of molecular helicity inversion controlled by twisting mode due to differences in the absolute configurations of the diamine moieties locked by the metal ions.

Colour: Pd yellow, C black, H white, N Purple (non-coordinated), red (*R*-configuration) and blue (*S*-configuration).

Corrected Conclusion section (page 16)

In this study, we have succeeded in selectively synthesising two twisted isomers of trinuclear Pd^{II}-macrocycles with markedly different rates of helicity inversion. In the tightly-twisted isomer **1_{tight}**, the three *ortho*-phenylenediamine moieties were folded inside the macrocyclic skeleton and the absolute configurations of the amine nitrogen atoms were *all-R* or *all-S*, while in the loosely-twisted isomers **1_{loose}**, the three *ortho*-phenylenediamine moieties were folded outside the skeleton and the absolute configurations of the amine nitrogen atoms were *alt-R/S*. In stark contrast to **1_{tight}**, which shows a very slow or no inversion, **1_{loose}** exhibits fast helicity inversion (1.38 s⁻¹ at 293 K in acetone-*d*₆). Moreover, the inversion kinetics can be controlled by isomerisation in a range from **1_{tight}**, where no inversion is detected, to **1_{loose}**, where inversion is fast. Our approach to the control of helicity inversion motion by the twisted isomers resulting from the configurational locking with metal ions is quite different from conventional approaches that require chemical substitutions or additives to control twisting motions. The new strategy of controlling molecular motions by the mode of twisting with coordinating atoms of different configuration is expected to be applicable to a variety of systems and can be expanded to the design of more sophisticated molecular machines.

Description newly added to the Supplementary Information (page S26)

2.4 Definition of the two twisted isomers of **1**

The two twisted isomers of **1** are defined by the degree to which the *ortho*-phenylenediamine moieties are folded inside the macrocycle. The degree of folding is evaluated by the dihedral angle between the plane horizontal to the macrocyclic skeleton and the plane of the *ortho*-phenylenediamine ring. The dihedral angle in **1** before metal coordination is nearly 0°, indicating no twisting. One isomer of the twisted Pd^{II}₃-macrocycles with a dihedral angle of greater than 90° is defined as the tightly-twisted isomer, **1_{tight}**. In contrast, the other isomer with a dihedral angle of less than 90° is therefore defined as the loosely-twisted isomer, **1_{loose}**. As a result of the different twisting modes, the absolute configuration of the amine nitrogen atoms changes through metal coordination to (*all-R* or *all-S*) and (*alt-R/S*) for the tightly- and loosely-twisted isomers, respectively. Note that this definition differs from the typical definition of a helix, which is based on differences in helical pitch. This is because the structure of **1** is not a helix, but a twist with the *P*- and *M*-helicity.

Figure S33. Illustration supporting the definition of the twisted isomers: (left) **L** is twisted tightly to form (*P*)-**1_{tight}** forming the (*all-R*) configuration with *ortho*-phenylenediamine nitrogen atoms, and (right) **L** is twisted loosely to form (*P*)-**1_{loose}** with the (*alt-R/S*) configuration with *ortho*-phenylenediamine nitrogen atoms.

2. As the authors mentioned, the "helicity inversion" of the tightly-twisted isomer requires inversion of the stereochemistry between the RR and SS configurations of the amine nitrogen atoms for each Pd center. This means that the RR configuration can be inverted into the SS form only via the RS/SR configuration if cleavage/rebound of the Pd-N bonds occurs in a stepwise manner. The loosely-twisted isomer having the *alt-R/S* configuration is the most probable intermediate during the possible helicity inversion of the tightly-twisted all-R and all-S forms, although this helicity inversion was not observed due to the relative thermodynamic stability of the loosely-twisted form. Therefore, the irreversible transformation to the loosely-twisted form is inevitable during the "helicity inversion" of the tightly-twisted forms, and the helicity inversion pathways of the two isomers cannot be separately discussed.

Our responses:

Thank you for the important comment, and we completely agree with this comment. According to the reviewers' comment, we have added a new discussion to clarify this point as follows. On the other hand, the fact that the helicity inversion pathways of the two isomers cannot be separately discussed is an important aspect that characterizes the twisted isomerism of this study and does not at all detract from the value of our strategy.

Correction to explanation in text (page 13) (Yellow parts: correction related to this comment)

Helicity inversion versus twist loosening observed in **1_{tight} (Fig. 6).** The helicity inversion rate was evaluated using (*M*)-enantio-enriched **1_{tight}**. An acetone-*d*₆ solution of the (*M*)-enantio-enriched **1_{tight}** (*P*:*M* = 37:63) was allowed to stand at 293 K for 3 days (Fig. 5g). After the solvent was removed, the enantiomeric ratio was examined using Δ -4 in CD₂Cl₂, and its enantiomeric ratio (*P*:*M* = 38:62) was almost the same as that of the starting material. This indicates that either the helicity inversion is too slow to be detected or that the inversion of **1_{tight}**

does not occur under this condition (Fig. 5f, Supplementary Sections 7.1–7.2). Similarly, the inversion rate in CD_2Cl_2 was also examined at 293 K, but no inversion was observed in 10 days (Supplementary Figs. 111 and 112). Besides, $\mathbf{1}_{\text{tight}}$ gradually isomerised to $\mathbf{1}_{\text{loose}}$ as described above, and the isomerisation to $\mathbf{1}_{\text{loose}}$ was observed during these analyses. These results suggest that the rate of isomerisation from $\mathbf{1}_{\text{tight}}$ to $\mathbf{1}_{\text{loose}}$ (twist loosening) ($5.7 \times 10^{-6} \text{ s}^{-1}$) is faster than that between (*P*)- and (*M*)- $\mathbf{1}_{\text{tight}}$ (helicity inversion). The faster isomerisation from $\mathbf{1}_{\text{tight}}$ to $\mathbf{1}_{\text{loose}}$ can be explained from the number of amine nitrogen atoms whose absolute configuration inverts. That is, in the case of (*all-R*) or (*all-S*) \rightarrow (*alt-R/S*) (twist loosening), only three of the six amine portions need to be inverted, but in the case of (*all-R*) \rightleftharpoons (*all-S*) (helicity inversion), all six amine portions must be inverted. Since configurational inversion of the amine moieties involves dissociation of the N–Pd or N–H bonds, the number of nitrogen inversion sites may affect the rate of the isomerisations. This consideration is also applied to understanding that the helicity inversion of $\mathbf{1}_{\text{loose}}$ (1.38 s^{-1} at 293 K, estimated by the Eyring plot, Supplementary Fig. 52) is much faster than that of $\mathbf{1}_{\text{tight}}$. This is because the helicity inversion of $\mathbf{1}_{\text{loose}}$ does not require the configurational changes of amine nitrogen atoms (*(alt-R/S) \rightleftharpoons (alt-R/S)*). On the other hand, the inversion of the *all-R* or *all-S* configuration in $\mathbf{1}_{\text{tight}}$ probably needs to occur in a stepwise manner via the intermediary *alt-R/S* configuration, but the intermediate corresponding to thermodynamically stable $\mathbf{1}_{\text{loose}}$ is no longer isomerised to the *all-S* or *all-R* configuration, respectively.

Reviewers' Comments:

Reviewer #1:

Remarks to the Author:

I see that in the revised manuscript the authors have carefully addressed all of the three reviewers' concerns. Now I am happy to see this beautiful work published as it is.

Reviewer #2:

Remarks to the Author:

I believe that the authors' have addressed the comments I made in my original review. In my opinion, this manuscript is now suitable for publication in Nature Communications.

Reviewer #3:

Remarks to the Author:

In this submission, the authors revised the manuscript according to the reviewers' comments.

However, the isomerization / inversion processes are still confusing and difficult for readers to understand. This is probably because the authors did not provide all the possible diastereomers of Pd3LCI6 structures.

The authors are strongly encouraged to solve this problem, for example, by providing an additional scheme in the main article figure showing all the diastereomers and the possible isomerization network pathways among all the possible isomers (In addition to the isomers mentioned in the article, there are many other isomers originating from the R/S isomerism):

(RR,RR,RR) / (SS,SS,SS) = "all-R" and "all-S"

(RR,RR,RS) / (SS,SS,RS)

(RR,RS,RS) / (SS,RS,RS)

(RS,RS,RS) = (RS,RS,RS) [meso] = "alt RS"

(RR,RR,SS) / (SS,SS,RR)

(RR,SS,RS) / (SS,RR,SR)

(SS,RR,RS) / (RR,SS,SR)

(RR,RS,RS) / (SS,RS,RS)

[Please check that the above list covers all the possible isomers]

This scheme should help readers to understand that only selected diastereomers, (all-R, all-S, alt-RS), appeared in this study, and that no R/S isomerization is required in the inversion of the loose isomer whereas many R/S isomerization steps would be necessary for (unobserved) P/M inversion of the tight isomers, "all-R" and "all-S".

Responses to the editor and reviewers' comments

The title of this paper has been revised as follows.

Original title: Selective synthesis of tightly- and loosely-twisted metallomacrocyclic isomers: sharp control of helicity inversion motion

Revised title: Selective synthesis of tightly- and loosely-twisted metallomacrocyclic isomers towards precise control of helicity inversion motion

Editor's comments and our responses

Dear Professor Shionoya,

Your manuscript entitled "Selective synthesis of tightly- and loosely-twisted metallomacrocyclic isomers: sharp control of helicity inversion motion" has now been seen again by our referees, whose comments appear below. In light of their advice I am delighted to say that we are happy, in principle, to publish a suitably revised version in Nature Communications under the open access CC BY license (Creative Commons Attribution 4.0 International License).

We therefore invite you to revise your paper one last time to address the remaining concerns of our reviewers and our editorial requests in the attached documents. At the same time we ask that you edit your manuscript to comply with our policies and formatting requirements and to maximise the accessibility and therefore the impact of your work.

Please note that it may still be possible for your paper to be published before the end of 2023, but in order to do this we will need you to address these points as quickly as possible so that we can move forward with your paper.

Please see the attached documents, listing a number of points that must be addressed. Failure to comply with our editorial requests will cause delays in accepting your manuscript. Please also see the *Nature Communications* [formatting instructions](https://www.nature.com/documents/ncomms-formatting-instructions.pdf) for further information.

Our responses:

We deeply appreciate your generous decision to accept our manuscript in principle, subject to appropriate revisions. We have revised the text, figures and Supplementary Information according to reviewer's comments and editorial requests in the Checklist. We believe that the revised version fully meets the reviewer's suggestions and the editorial requests.

Reviewer #1's comments and our responses

I see that in the revised manuscript the authors have carefully addressed all of the three reviewers' concerns. Now I am happy to see this beautiful work published as it is.

Our responses:

We would like to express our deepest gratitude to the author for carefully reviewing our manuscript and providing valuable comments until the final version.

Reviewer #2's comments and our responses

I believe that the authors' have addressed the comments I made in my original review. In my opinion, this manuscript is now suitable for publication in Nature Communications.

Our responses:

We would like to express our deepest gratitude to the author for carefully reviewing our manuscript and providing valuable comments until the final version.

Reviewer #3's comments and our responses

In this submission, the authors revised the manuscript according to the reviewers' comments. However, the isomerization / inversion processes are still confusing and difficult for readers to understand. This is probably because the authors did not provide all the possible diastereomers of Pd3LCI6 structures.

The authors are strongly encouraged to solve this problem, for example, by providing an additional scheme in the main article figure showing all the diastereomers and the possible isomerization network pathways among all the possible isomers (In addition to the isomers mentioned in the article, there are many other isomers originating from the R/S isomerism):

(RR,RR,RR) / (SS,SS,SS) = "all-R" and "all-S"

(RR,RR,RS) / (SS,SS,RS)

(RR,RS,RS) / (SS,RS,RS)

(RS,RS,RS) = (RS,RS,RS) [meso] = "alt RS"

(RR,RR,SS) / (SS,SS,RR)

(RR,SS,RS) / (SS,RR,SR)

(SS,RR,RS) / (RR,SS,SR)

(RR,RS,RS) / (SS,RS,RS)

[Please check that the above list covers all the possible isomers]

This scheme should help readers to understand that only selected diastereomers, (all-R, all-S, alt-RS), appeared in this study, and that no R/S isomerization is required in the inversion of the loose isomer whereas many R/S isomerization steps would be necessary for (unobserved) P/M inversion of the tight isomers, "all-R" and "all-S".

Our responses:

Thank you for your helpful suggestion. As you pointed out, the explanation regarding the isomerisation and inversion processes of **I_{tight}** may still confuse readers to some extent. Therefore, we have added a scheme showing all possible diastereomers and isomerisation pathways between them as Fig. 7. This figure is believed to help readers more easily understand the isomerisation and inversion processes discussed in this article. Fig.7 and its caption are newly added to the text.

Fig. 7| Possible isomers and their isomerisation pathways of 1. All possible diastereomers and their isomerisation pathways for **1** are shown in this scheme. The black circles indicate the helically twisted macrocyclic structures of **1**. The notions *RR*, *SS* and *SR* above the black circles indicate pairs of the absolute configurations of the amine nitrogen atoms of the *ortho*-phenylenediamine moiety observed in the crystal strictures. The isomerisation pathways to the *RS* configuration are excluded in this scheme, as a pair of the amine nitrogen atoms with the *RS* configurations, with the two amine protons pointing towards the inside of the macrocycle, was not observed experimentally. The green arrows indicate the direct isomerisation pathways from **1_{tight}** to **1_{loose}**, and the dotted arrows indicate other possible isomerisation pathways.